



# On the drivers of droplet variability in Alpine mixed-phase clouds

Paraskevi Georgakaki[1], Aikaterini Bougiatioti[2], Jörg Wieder[3], Claudia Mignani[4], Fabiola Ramelli[3], Zamin A. Kanji[3], Jan Henneberger[3], Maxime Hervo[5], Alexis Berne[6], Ulrike Lohmann[3] and Athanasios Nenes[1,7]

[1]Laboratory of Atmospheric Processes and their Impacts, School of Architecture, Civil & Environmental Engineering, École Polytechnique Fédérale de Lausanne, Lausanne, CH-1015, Switzerland
[2]Institute for Environmental Research & Sustainable Development, National Observatory of Athens, P. Penteli, GR-15236, Greece
[3]Department of Environmental Systems Science, Institute for Atmospheric and Climate Science, ETH Zurich, Zurich, CH-8092, Switzerland
[4]Department of Environmental Sciences, University of Basel, Basel, CH-4056, Switzerland
[5]Federal Office of Meteorology and Climatology, MeteoSwiss, Payerne, CH-1530, Switzerland
[6]Environmental Remote Sensing Laboratory, School of Architecture, Civil & Environmental Engineering, École Polytechnique Fédérale de Lausanne, Lausanne, CH-1015, Switzerland
[7]Center for Studies of Air Quality and Climate Change, Institute of Chemical Engineering Sciences, Foundation for Research and Technology Hellas, Patras, GR-26504, Greece

*Correspondence to*: Athanasios Nenes (athanasios.nenes@epfl.ch).

**Abstract**

Droplet formation provides a direct microphysical link between aerosols and clouds (liquid or mixed phase), and its adequate description poses a major challenge for any atmospheric model. Observations are critical for evaluating and constraining the process. Towards this, aerosol size distributions, cloud condensation nuclei, hygroscopicity and lidar-derived vertical velocities were observed in Alpine mixed-phase clouds during the Role of Aerosols and Clouds Enhanced by Topography on Snow (RACLETS) field campaign in the Davos, Switzerland region during February and March 2019. Data from the mountain-top site of Weissfluhjoch (WFJ) and the valley site of Davos Wolfgang are studied. These observations are coupled with a state-of-the art droplet activation parameterization to investigate the aerosol-cloud droplet link in mixed-phase clouds. The mean CCN-derived hygroscopicity parameter, $\kappa$, at WFJ ranges between 0.2-0.3, consistent with expectations for continental aerosol. $\kappa$ tends to decrease with size, possibly from an enrichment in organic material associated with the vertical transport of fresh ultrafine particle emissions (likely from biomass burning) from the valley floor in Davos. The parameterization provides droplet number that agrees with observations to within ~25%. We also find that the susceptibility of droplet formation to aerosol concentration and vertical velocity variations can be appropriately described as a function of the standard deviation of the distribution of updraft velocities, $\sigma_w$, as the droplet number never exceeds a characteristic limit, termed "limiting droplet number", of ~150-550 cm$^{-3}$, which depends solely on $\sigma_w$. We also show that high aerosol levels in the valley, most likely from anthropogenic activities, increase





cloud droplet number, reduce cloud supersaturation (<0.1%) and shift the clouds to a state that
is less susceptible to aerosol and become very sensitive to vertical velocity variations. The
transition from aerosol to velocity-limited regime depends on the ratio of cloud droplet number
to the limiting droplet number, as droplet formation becomes velocity-limited when this ratio
exceeds 0.5. Under such conditions, droplet size tends to be minimal, reducing the likelihood
that large drops are present that promote glaciation through rime splintering and droplet
shattering. Identifying regimes where droplet number variability is dominated by dynamical –
rather than aerosol – changes is key for interpreting and constraining when and which types of
aerosol effects on clouds are active.

## 1. Introduction

Orographic clouds, and the precipitation they generate, play a major role in Alpine weather and
climate (e.g., Roe, 2005; Grubisic and Billings, 2008; Saleeby et al., 2013; Vosper et al., 2013;
Lloyd et al., 2015). The formation and evolution of orographic clouds involves a rich set of
interactions at different spatial and temporal scales encompassing fluid dynamics, cloud
microphysics and orography (Roe, 2005; Rotunno and Houze, 2007). Atmospheric aerosol
particles modulate the microphysical characteristics of orographic clouds by serving as cloud
condensation nuclei (CCN) that form droplets, or ice nucleating particles (INPs) that form ice
crystals (e.g., Pruppacher and Klett, 1997; Muhlbauer and Lohmann, 2009; Zubler et al., 2011;
Saleeby et al., 2013).
Emissions of aerosol particles acting as CCN and INPs can affect the microphysical and
radiative properties of clouds with strong (but highly uncertain) effects on local and regional
climate (IPCC, 2013; Seinfeld et al., 2016). Aerosol interactions with orographic clouds are
subject to even larger uncertainties, owing in part to the complex flows generated by the
interaction of the large-scale flow with the mesoscale orographic lifting and condensation, and
complex anisotropic turbulent air motions that arise (Roe, 2005; Smith, 2006; Rotunno and
Houze, 2007). Most importantly, orographic clouds are often mixed-phase clouds (MPCs),
which are characterized by the simultaneous presence of supercooled liquid water droplets and
ice crystals (Lloyd et al., 2015; Farrington et al., 2016; Lohmann et al., 2016; Henneberg et al.,
2017). MPCs remain one of the least understood cloud types, due to the multiple and highly
nonlinear cloud microphysical pathways that can affect their properties and evolution. MPCs
tend to glaciate (i.e., transition to pure ice clouds) over time because of the Bergeron-Findeisen
process, which is the rapid growth of ice crystals at the expense of the evaporating cloud



droplets, owing to the higher saturation vapor pressure of liquid water over ice (Bergeron,
1935; Findeisen, 1938). Aerosol concentrations may also alter the microphysical pathways
active in MPCs and ultimately drive their glaciation state. For instance, increase in CCN
concentrations leads to more numerous and smaller cloud droplets, reducing the riming
efficiency of ice crystals and therefore the hydrometeor crystal mass and the amount of
precipitation (Lohmann and Feichter, 2005; Lance et al., 2011; Lohmann, 2017). This
mechanism counters the glaciation indirect effect, where increases in INP concentrations
elevate ice crystal number concentration (ICNC) and promotes the conversion of liquid water
to ice - therefore the amount of ice-phase precipitation (Lohmann, 2002). Increases in CCN
can also decrease cloud droplet radius, and impede cloud glaciation, owing to reductions in
secondary ice production (SIP), which includes rime splintering, collisional break-up and
droplet shattering (Field et al., 2017; Sotiropoulou et al., 2020a, 2020b).
Cloud-scale updraft velocity (i.e., the part of the vertical velocity spectrum with positive
values) is the major driver of droplet formation, owing to the supersaturation generated from
adiabatic expansion and cooling (e.g., Nenes et al., 2001; Ghan et al., 2011). Despite its
importance, the simulation of updraft velocity by atmospheric models is rarely constrained by
observations, which can lead to large uncertainties in climate and numerical weather prediction
models (Sullivan et al., 2016, 2018). Reutter et al. (2009) pointed out that droplet formation in
clouds can be limited by the amount of CCN present (called the "aerosol-limited" regime), or
the vertical velocity that generates supersaturation in the cloudy updrafts (called the "velocity-
limited" regime). Over the complex Alpine terrain, vertical motions can be significantly shaped
by the effects of orography (Lohmann et al., 2016). Orographic MPCs have been frequently
observed in the Swiss Alps under high updraft velocity conditions, where supersaturation with
respect to liquid water is formed faster than it is depleted by diffusional and collisional ice
growth processes (Korolev and Isaac, 2003) leading to persistent MPCs (Lohmann et al., 2016).
Given the importance of droplet number for the radiative cloud properties and
microphysical evolution of Alpine MPCs, it is essential to understand the main aerosol and
dynamics properties that drive droplet formation. A limited number of studies exist that discuss
this very important topic, focusing though on liquid-phase clouds (Hammer et al., 2014, 2015;
Hoyle et al., 2016). Hoyle et al. (2016) demonstrated that 79% of the variance in droplet number
observed in warm tropospheric clouds formed over the high-altitude research station of
Jungfraujoch (3450 m a.s.l.) in the Swiss Alps can be explained by the potential CCN number
concentrations (i.e. aerosol particles with a dry diameter >80 nm). With box model simulations,
Hammer et al. (2015) investigated the influence of updraft velocity, particle concentration and





hygroscopicity on droplet formation in cloud updraft, and found that variations in vertical wind
velocity have the strongest influence on aerosol activation. The ability to predict droplet
number in MPCs, where the existence of ice crystals can deplete supersaturation or the low
temperatures may decrease CCN activity through the formation of glassy aerosol, has not been
assessed in a closure study to date.

Here we analyze observational data collected as part of the Role of Aerosols and Clouds

Enhanced by Topography on Snow (RACLETS) field campaign, which was held in the region
of Davos, Switzerland, during February and March 2019. This intensive field campaign aims
to address questions related to the modulators of orographic precipitation, the drivers of the
enhanced ice-crystal number concentrations observed in MPCs as well as the human-caused
pollution effects on cloud microphysical and optical properties. Through this study we focus
on a two-week period seeking to unravel the complex aerosol-droplet-updraft velocity
interactions that occur in the orographic MPCs. For this, we combine CCN number
concentrations with the particle size distributions to understand the variations in hygroscopicity
over time and for sites located in the valley and a close by mountain-top site. The in-situ
measurements are subsequently coupled with a state-of-the art droplet parameterization to
determine the potential droplet numbers and the corresponding maximum supersaturation
achieved in cloudy updrafts. The predicted droplet numbers are evaluated against direct
observations, and the degree to which droplet formation is velocity- or aerosol-limited is
determined for the whole timeseries.

## 2.    Methods

2.1 Observational datasets
The analysis utilizes measurements collected during the RACLETS campaign, which took
place from 8 February to 28 March 2019 (https://www.envidat.ch/group/about/raclets-field-
campaign) (Mignani et al., 2020; Ramelli et al., 2020b, c; Lauber et al., 2020). This joint
research project offers a unique dataset of orographic clouds, precipitation and snow
measurements in an effort to shed light on some fundamental microphysical processes being
present in subsequent stages of the lifecycle of clouds (i.e. cloud formation, precipitation onset,
cloud dissipation). All measurements presented in this paper were performed at two distinct
observation stations near Davos, Switzerland (supplement Fig. S1). A measurement site is
located at Davos Wolfgang, which is the pass between Davos (1560 m a.s.l.) in the South and
Klosters (1200 m a.s.l.) in the North and is otherwise known as Wolfgang-Pass (WOP; 1630
m a.s.l., 46°50'08.076″N 9°51'12.939″E). Measurements were also conducted at the mountain-
top station Weissfluhjoch (WFJ; 2700 m a.s.l., 46°49'58.670″N 9°48'23.309″E), which is
located ⌣1 km above the valley floor in Davos, in the eastern part of the Swiss Alps. The
current study primarily focuses on data collected during a two-week period of interest, which
spans from 24 February to 8 March 2019. During the RACLETS campaign, a defective sheath
air filter affected the CCN measurements collected at WFJ, thus inhibiting data usage from the
instrument for a large duration of the campaign. Therefore, we limit our analysis to the above-
mentioned period, when the CCN counter was fully operational. Besides, during the selected
period two distinct weather patterns were observed (fair weather conditions interrupted by a
precipitating period), allowing for a contrasting analysis of the observed scenarios. The
following description refers to the measurements that provided the basis for the present analysis
(see Table 1).

*2.1.1 Aerosol particle size distribution measurements*
Particle size distributions were continuously monitored at WOP and WFJ using commercially
available Scanning Mobility Particle Sizers (SMPS; Model 3938, TSI Inc., US). At both
stations, the systems consisted of a differential mobility analyzer (Model 3081, TSI Inc., US),
a soft X-ray neutralizer (Model 3088, TSI Inc., US) and a water-based condensation particle
counter (Model 3787 at WOP, Model 3788 at WFJ, TSI Inc. US). Running the particle counters
in low flow mode ($0.6\ \mathrm{Lmin}^{-1}$), using a sheath flow of $5.4\ \mathrm{Lmin}^{-1}$ and applying a total scanning
time of 2 minutes (scan time: 97 s, retrace time: 3 s, purge time: 10 s), particle size distributions
between 11.5 nm and 469.8 nm were monitored.

*2.1.2 CCN measurements*
A Droplet Measurement Technologies (DMT) single-column continuous-flow streamwise
thermal gradient chamber (CFSTGC; Roberts and Nenes, 2005) was used to carry out in-situ
measurements of CCN number concentrations for different supersaturations (*SS*). The
CFSTGC consists of a cylindrical flow tube with wetted walls, inside which *SS* is developed
by applying a linear streamwise temperature gradient between the column top and bottom.
Owing to the greater mass diffusivity of water vapor than the thermal diffusivity of air, a
constant and controlled *SS* is generated with a maximum at the centerline of the flow tube. The
*SS* is mainly dependent on the applied temperature gradient, flow rate and pressure (Roberts
and Nenes, 2005). An aerosol sample flow is introduced at the column centerline, and those
particles having a critical supersaturation lower than the instrument *SS* will activate to form



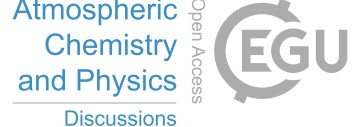

droplets and will afterward be counted and sized by an Optical Particle Counter (OPC) located
at the base of the CFSTGC column. The *SS* developed within the instrument responds linearly
to changes in pressure, since its operation relies on the difference between heat and mass
diffusivity. Calibration of the instrument, which determines the output supersaturation, was
performed by the manufacturer at ~800 mbar, while throughout the campaign the CFSTGC
was operating at a lower pressure ~735 mbar, therefore the *SS* reported by the instrument is
adjusted by a factor of $\frac{735}{800} = 0.92$, which takes into account the difference between the
ambient and the calibration pressure (Roberts and Nenes, 2005). CCN concentrations were
measured at a specific *SS* for approximately 10 minutes; the instrument was cycled between 6
discrete values ranging from 0.09% to 0.74% supersaturations, producing a full spectrum every
hour. Each 10-minute segment of the raw CCN data are filtered to discount periods of transient
operation (during supersaturation changes), and whenever the room temperature housing the
instrument changed sufficiently to induce a reset in column temperature (the instrument control
software always sets the column temperature to be at least 1.5 degrees above the room
temperature to exclude spurious supersaturation generation in the column inlet). The CFSTGC
was deployed on the mountain-top site of WFJ with the intention of relating the CCN
measurements directly to the size distribution and total aerosol concentration data measured by
the SMPS instrument at the same station.

*2.1.3 Cloud microphysical measurements*
In-situ observations of the cloud microphysical properties were obtained with the tethered
balloon system HoloBalloon (Ramelli et al., 2020a). The main component of the measurement
platform is the holographic cloud imager HOLIMO 3B, which uses digital in-line holography
to image an ensemble of cloud particles in the size range from 6 μm to 2 mm diameter in a
three-dimensional detection volume. Based on a set of two-dimensional images, information
about the particle position, size and shape can be obtained. The detected particles can be
classified as cloud droplets and ice crystals using supervised machine learning (Fugal et al.,
2009; Touloupas et al., 2020). The differentiation is possible for particles larger than 25 μm.
From the classification, the phase-resolved size distribution, concentration and content can be
derived (Henneberger et al., 2013; Ramelli et al., 2020a). HOLIMO has an open path
configuration (i.e. the detection volume lies between the two instrument towers) and thus is
also able to measure raindrops up to a size of ⌣2 mm. The HoloBalloon platform was flying at
WOP and provided vertical profiles of the cloud properties within the lowest 300 meters of the





boundary layer (BL). The current analysis utilizes the cloud droplet number concentration and
liquid water content (LWC) measurements. Note that the LWC is calculated based on the
measured number concentration and size distribution using a water density of 1000 kg m$^{-3}$ and
is therefore dominated by large cloud particles.

**Table 1.** Overview of data sources from the RACLETS campaign used for this study. Along
with the observed parameters, the corresponding instrumentation, measurements range and
time resolutions are listed.

| Measured parameter | Measurement site | Instrument | Measurement range | Time resolution |
|---|---|---|---|---|
| Aerosol number size distribution | WOP/ WFJ | Scanning Mobility Particle Sizer | 11.5 – 469.8 nm | 2 min |
| CCN number concentration | WFJ | Continuous flow streamwise thermal gradient CCN counter | $SS = 0.09 – 0.74\%$ | 1 s |
| Cloud droplet number concentration and liquid water content | WOP | Holographic cloud imager HOLIMO | 6 µm – 2 mm | 10 – 20 s |
| Precipitation | WOP/ WFJ | Parsivel disdrometer/ MeteoSwiss weather station | 0.2 mm – 25 mm | 30 s |
| Horizontal wind speed and direction | WOP/ WFJ | MeteoSwiss weather station | – | 10-min averages |
| Profiles of vertical wind speed | WOP | Wind Doppler Lidar | 200 m – 8100 m AGL | up to 5 s |


*2.1.4 Meteorological data*
During the measurement period, meteorological parameters (e.g., pressure, temperature,
precipitation, horizontal wind speed and direction) were continuously monitored by the
permanent MeteoSwiss observation station at WFJ. Additionally, a weather station was
installed on the OceaNet container (Griesche et al., 2019) deployed at WOP, which also hosted





several remote sensing instruments (e.g., Cloud radar, Raman Lidar, Microwave radiometer)
and a Particle Size Velocity (Parsivel) disdrometer (Parsivel2, OTT HydroMet GmbH,
Germany; Tokay et al., 2014) to measure precipitation. As there was no wind sensor included
in the weather station on the OceaNet container, we utilized the horizontal wind speed and
direction measurements from the nearby MeteoSwiss station in Davos, assuming that they
provide a good proxy for the wind regime in the valley. Vertical wind speed profiles were
obtained with a wind Doppler Lidar (WindCube 100S, manufactured by Leosphere) at WOP.
Throughout the campaign the wind lidar measured from 200 m to 8100 m above ground level
(AGL) with high temporal (up to 5 s) and vertical resolution (50 m). The wind lidar operated
following the Doppler Beam Switching (DBS) technique with an elevation of 75˚. More
information about the remote sensing measurements can be found in Ramelli et al. (2020b).

2.2 Aerosol hygroscopicity
The aerosol hygroscopicity parameter, $\kappa$, encompasses the impact of particle chemical
composition on its subsaturated water uptake and CCN activity (Petters and Kreidenweis,
2007). Here, we determine $\kappa$ similar to the approach of Moore et al. (2011), Jurányi et al.
(2011), Lathem et al. (2013), Kalkavouras et al. (2019), Kacarab et al. (2020) and others, by
combining the CCN measurements with the SMPS aerosol size distribution data as follows.
For each SMPS scan, the particle size distribution is integrated backward starting from the bin
with the largest-size particles – which corresponds to CCN with the lowest critical
supersaturation, $S_{cr}$. We then successively add bins with smaller and smaller diameters, until
the aerosol number matches the CCN concentration observed for the same time period as the
SMPS scan. The particles in the smallest size bin, which we call *critical dry diameter*, $D_{cr}$,
correspond to CCN with highest critical supersaturation possible – being the instrument
supersaturation, $SS$. From $D_{cr}$ and $SS$ we determine $\kappa$ from Köhler theory (Petters and
Kreidenweis, 2007), assuming the particle chemical composition is internally mixed:

$$\kappa = \frac{4A^3}{27D_{cr}^3 SS^2} \tag{1}$$

where $A = \frac{4M_w \sigma_w}{RT\rho_w}$ is the Kelvin parameter, while $M_w$ (kg mol$^{-1}$), $\sigma_w$ (J m$^{-2}$) and $\rho_w$ (kg m$^{-3}$) are,
respectively, the molar mass, surface tension and density of water, $R$=8.3145 J mol$^{-1}$ K$^{-1}$ is the
universal gas constant and $T$ (K) is the ambient temperature. The $\kappa$ determined above represents
the composition of particles with diameter $D_{cr}$ (large particles can have a different $\kappa$ but still
activate given that their critical supersaturation is lower than the prevailing $SS$ in the CCN





chamber). This means that over the course of an hour, over which a full *SS* cycle is completed,
$\kappa$ is determined for a range of $D_{cr}$, which in our case were in the range of 50-200 nm (Section
3.1). This size-resolved $\kappa$ information provides insights on the possible origin and chemical
components of the aerosol, which is important given that there is no other measurement
available to constrain chemical composition during RACLETS. From $\kappa$, we infer an equivalent
organic mass fraction, $\varepsilon_{org}$, assuming that the aerosol is composed of an organic-inorganic
mixture:

$$\varepsilon_{org} = \frac{(\kappa - \kappa_i)}{(\kappa_o - \kappa_i)} \tag{2}$$

where $\kappa_i = 0.6$ and $\kappa_o = 0.1$ are characteristic hygroscopicity values for the inorganic fraction
of aerosol (represented by ammonium sulphate), and aged organics, respectively (Petters and
Kreidenweis, 2007; Wang et al., 2008; Dusek et al., 2010). Note that these values for a
continental aerosol are supported by observations and analyses (e.g., Andreae and Rosenfeld,
2008; Rose et al., 2008; Pringle et al., 2010).

2.3 Cloud droplet number and cloud maximum supersaturation
Here we apply adiabatic cloud parcel theory to the observational datasets to determine the
maximum in-cloud supersaturation ($S_{max}$) and cloud droplet number ($N_d$) that would form over
both measurement sites throughout the observation period. Droplet calculations are carried out
with the physically based aerosol activation parameterization of Nenes and Seinfeld (2003),
with extensions introduced by Fountoukis and Nenes (2005), Barahona et al. (2010), and
Morales and Nenes (2014). Each $N_d$ calculation requires knowledge of the observed pressure,
temperature, vertical winds, aerosol size distribution and hygroscopicity. For the WFJ site, all
data are available as described in the sections above. For the WOP site, CCN (hence
hygroscopicity) data are not available, so we carry out $N_d$ calculations at two $\kappa$ values, 0.1 and
0.25, which is the upper and the lower limit determined from the WFJ analysis (Section 3.1).
The ability to reproduce observed cloud droplet number concentrations ("Method evaluation",
Section 3.2.1) further supports the selection of these values.

Vertical velocity measurements are obtained from the wind lidar data extracted for the

altitude of interest, being 200 m and 1100 m AGL for WOP and WFJ, respectively. The
measured updraft velocities are then fitted to a half-Gaussian probability density function
(PDF) with zero mean and standard deviation $\sigma_w$. PDFs are obtained for hourly segments, while
an example of this calculation method is provided in the supplementary material (supplement



Fig. S2). Employing the "characteristic velocity" approach of Morales and Nenes (2010), the
PDF-averaged values of $N_d$ and $S_{max}$ are calculated by applying the parameterization using a
single characteristic velocity, $w^*=0.79\sigma_w$. This approach has been shown to successfully predict
cloud-scale values of $N_d$ in field studies for cumulus and stratocumulus clouds (e.g., Conant et
al., 2004; Meskhidze et al., 2005; Fountoukis et al., 2007; Morales et al., 2014; Kacarab et al.,
2020). The droplet closure carried out in this study is also used to support the validity of this
approach for Alpine MPCs. The accuracy of the wind lidar products is affected by precipitation,
as the measured updraft velocities might be masked by the terminal fall velocity of the
hydrometeors. We therefore exclude precipitating periods from our analysis – using
disdrometer measurements to constrain periods of precipitation. Aiming to examine how $N_d$
responds to different vertical velocity-aerosol situations, as a sensitivity test, potential $N_d$ for
both sites are calculated at values of $\sigma_w$ between 0.1 and 1.0 ms$^{-1}$ that cover the observed range
(Section 3.2.4). Note that we use the term "potential" droplet number throughout this study, as
its calculation is performed regardless of the actual existence of clouds over the measurement
sites.

**3.    Results & Discussion**
3.1 Particle number, CCN concentration and $\kappa$ at WOP and WFJ
The total aerosol number concentration ($N_{aer}$) timeseries (integrated aerosol size distribution)
together with horizontal wind speed and direction measurements are depicted for both sites in
Figure 1. The $N_{aer}$ data points of WFJ are colored by $\kappa$ (Section 2.2), while orange dots are
used as a trace for WOP timeseries, as $\kappa$ was not determined for the site owing to a lack of
corresponding CCN measurements. Aiming to interpret the aerosol variations and the potential
differences observed between valley and high-altitude measurements, the two-week period of
interest is divided into two different sub-periods. During 24 and 28 of February, a high-pressure
system was dominant over Europe with clear skies and elevated temperatures. During this first
sub-period, the $N_{aer}$ varies considerably, and tends to follow a diurnal cycle that anticorrelates
between the two sites (Fig. 1a). The concentrations at WOP are most of the times elevated with
respect to WFJ, which is expected as the $N_{aer}$ in the valley is higher – being influenced by local
sources, which during this time of the year include emissions from biomass burning (BB) (Lanz
et al., 2010). $N_{aer}$ at WOP peaks in the evening, reaching up to ~$10^4$ cm$^{-3}$ presumably because
of BB emissions in the valley which seem to stop around midnight (Fig. 1a). Up to 2 orders of
magnitude lower $N_{aer}$ is measured at the same time at the WFJ site. In the afternoon, aerosol



numbers at WFJ approach those observed at WOP, indicating that the two sites are possibly
experiencing similar air masses. The $\kappa$ for WFJ seems to follow a clear temporal pattern as
well, ranging between ∽0.1-0.4 with a minimum in the afternoon, when the two sites
experience the same air masses. Low $N_{aer}$ values are accompanied by higher $\kappa$, while at higher
$N_{aer}$ conditions less hygroscopic aerosols are recorded (Fig. 1a).

The above diurnal cycles and their relationships can be understood in terms of boundary

layer dynamics typically occurring in mountain valley systems (Chow et al., 2013). During
daytime, under clear sky conditions, the slopes and the valley itself are warmed by solar
radiation, causing rising of the BL, and additionally the production of buoyant air masses that
rise up the slope toward the summit (through "up-slope" and "up-valley" winds) (Okamoto and
Tanimoto, 2016). This hypothesis can be further supported by the fair weather recorded by the
weather station at WFJ until 28 February. The buoyant upslope flow could then transport
polluted air masses originating from the BL of the valley up to the WFJ site, elevating the
concentrations of less hygroscopic aerosols observed in the afternoon. The situation reverses
during nighttime, when cold air descends from the slopes (down-slope winds) and flows out of
the valley (down-valley winds) due to the radiative cooling of the surface. The less polluted air
observed during the early hours of the day before sunrise indicates that the WFJ station
remained in the free troposphere (FT), with lower $N_{aer}$ and more aged air (i.e. larger $\kappa$) with a
more prominent accumulation mode (Seinfeld and Pandis, 2006).

Another consideration is that the upslope flow that "connects" the valley and the

mountain-top site may not only be driven by thermal convection but also from mechanically-
forced lifting. The latter mechanism is caused by the deflection of strong winds by a steep
mountain slope and it can be of great importance depending mainly on the height of the
mountain and the mean speed of the wind (Kleissl et al., 2007). The local wind effects can be
further interpreted looking at the MeteoSwiss timeseries of wind speed and direction for both
stations (Fig. 1b, c). Wind measurements at WFJ station recorded a strong wind speed reaching
up to ∽11 ms$^{-1}$ from easterly-northeasterly directions between 24 and 28 of February. The wind
direction measured at WFJ coincides with the relative location of WOP site (see black dashed
line in Fig. 1c). The steep orography over the Alps would transform part of this strong
horizontal motion into vertical motion, and transport air from WOP to WFJ, as seen in other
Alpine locations, like Jungfraujoch (e.g., Hoyle et al., 2016). A detailed analysis however is
out of the scope of this study.

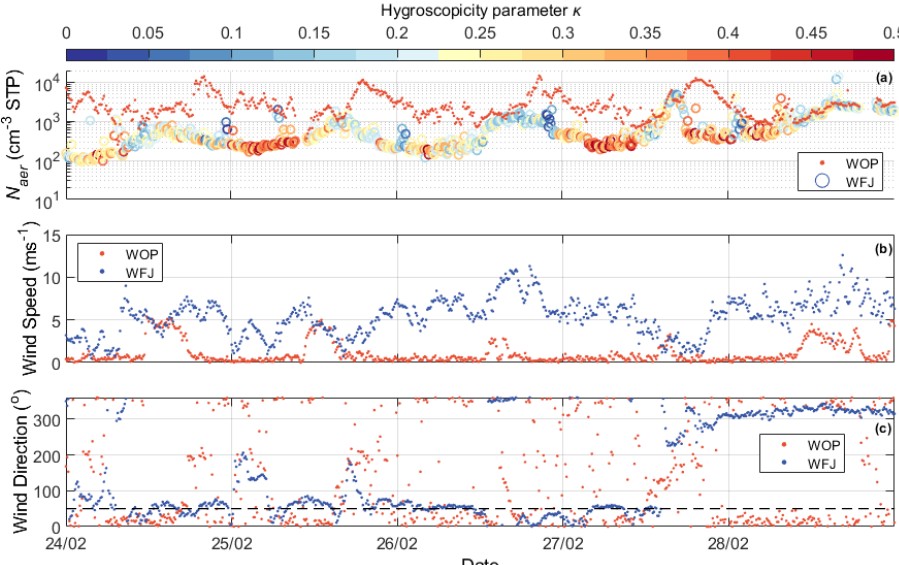


**Figure 1.** (a) $N_{aer}$ in standard temperature and pressure conditions (cm$^{-3}$ STP) at WOP (orange
dots) and at WFJ (circles colored by $\kappa$), (b) wind speed (ms$^{-1}$), and (c) wind direction (in
degrees) obtained from the MeteoSwiss observation stations at WFJ and Davos between 24
and 28 February 2019. The black dashed line indicates the relative direction of WOP to WFJ.
Each day is referenced to 00 UTC.


Focusing on the period between 1 and 8 March, meteorological observations show the

pressure and temperature dropping together with intense snow and rain events, associated with
the passage of cold fronts over the region. Three precipitation events are visible in our dataset
occurring on the 1$^{st}$, 4$^{th}$ and 7$^{th}$ of March 2019 creating up to ⁓7 mm per hour of precipitation
(Fig. 2). The most intense drop in $N_{aer}$ is seen to occur during and after the precipitation events,
with the aerosol concentrations dropping to less than 200 cm$^{-3}$ (100 cm$^{-3}$) at WOP (WFJ). This
is not the case for the last event, where a big "spike" of $N_{aer}$ is observed before the precipitation
event in WOP timeseries, which is in contrast with the concurrent sharp decrease in $N_{aer}$ (< 20
cm$^{-3}$) observed at WFJ. This could be an indication of a local source affecting the $N_{aer}$ recorded
in the valley. During dry weather conditions, we can notice again the aerosol timeseries
correlating during the afternoon and anticorrelating later in the evening-early morning hours.
On the 3$^{rd}$ of March, a steep increase in $N_{aer}$ is seen in WFJ timeseries reaching up to ⁓4000
cm$^{-3}$, which is followed by a period of several hours with low hygroscopicity values ($\kappa$<0.2)
indicating once more the influence of freshly emitted particles arriving at WFJ from the BL of
lower altitudes. Additionally, between 1 and 8 March, the diurnal cycle of particle





hygroscopicity is less pronounced. Especially on 1$^{st}$ and 7$^{th}$ of March nucleation processes or
precipitation scavenging removes the more hygroscopic aerosols from WFJ, thus leaving
behind the less effective CCN particles characterized by lower $\kappa$ values ($< 0.1$). Also, because
$N_{aer}$ drops, fresh local emissions become more important, further justifying the predominance
of low hygroscopicity values.

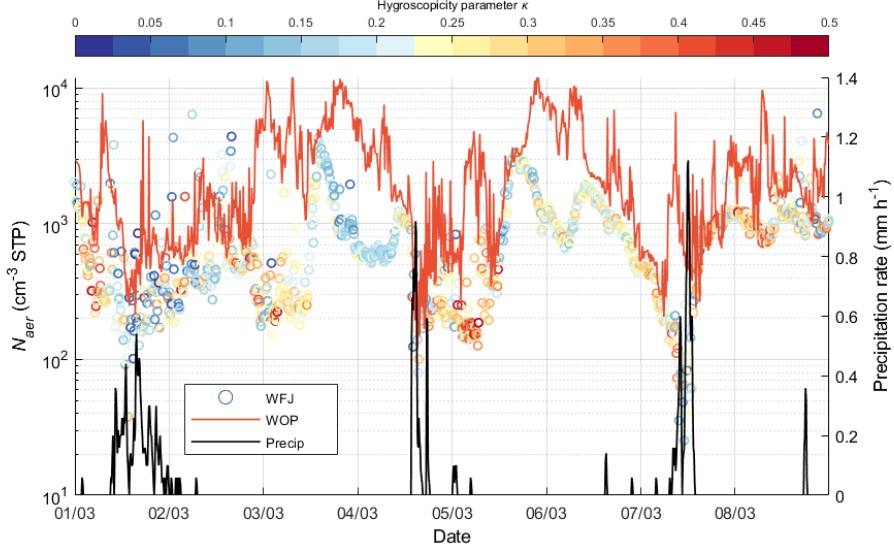


**Figure 2.** $N_{aer}$ (cm$^{-3}$ STP) at WOP (orange solid line) and at WFJ (circles colored by $\kappa$). The
black solid line represents the precipitation rate (mm h$^{-1}$) recorded from the MeteoSwiss
observation station for each 10-min interval at WFJ between 1 and 8 of March 2019.


Figure 3 presents the CCN number concentration timeseries at WFJ for all 6

supersaturations measured. Throughout the two-week measurement period the recorded CCN
number concentrations do not seem to follow a clear temporal pattern. The absence of a diurnal
cycle in CCN properties measured at Jungfraujoch during winter was also pointed out in the
study of Jurányi et al. (2011), because the site is mainly in free tropospheric conditions during
most of the winter. According to Figure 3, the observed CCN concentrations tend to be low
($\sim 10^2$ cm$^{-3}$) even at the highest $SS$ (0.74%), which is expected given that WFJ is a remote
continental measurement site with CCN concentrations that are typical of FT continental air
(Jurányi et al., 2010, 2011; Hoyle et al., 2016; Fanourgakis et al., 2019). This is again in line
with the measured monthly median values of CCN (at $SS$=0.71%) reported by Jurányi et al.
(2011) being equal to 79.1 and 143.4 cm$^{-3}$ for February and March 2009, respectively. Some
local CCN spikes are however recorded during the evening of 28 February and at the beginning



of March (e.g., on $2^{nd}$, $4^{th}$ and $6^{th}$ March), with the observed values of CCN reaching up to 650
$cm^{-3}$ at SS=0.09% (lowest SS) and 1361 $cm^{-3}$ at SS=0.74% (highest SS). Considering that WFJ
is a site frequently located in the FT, sudden fluctuations in the CCN concentrations could be
related to the vertical transport of freshly emitted particles (e.g., wood burning or vehicle
emissions) from the valley floor in Davos. It is also worthy to note that some aerosol spikes
observed on the $3^{rd}$ (∽ 3350 $cm^{-3}$) and the $5^{th}$ of March (∽ 2100 $cm^{-3}$) in the WFJ timeseries
(Fig. 1a) are not accompanied by a corresponding peak in the CCN timeseries. This indicates
the presence of small aerosol particles, which activate above 0.74% supersaturation (i.e.
particles with a diameter smaller than approx. 25 nm). This event could also be associated with
new particle formation (NFP) events. A previous study by Herrmann et al. (2015) reported the
aerosol number size distribution at the Jungfraujoch over a 6-year period indicating that NPF
was observed during 14.5% of the time without a seasonal preference. Tröstl et al. (2016) also
showed that NPF significantly adds to the total aerosol concentration at Jungfraujoch and is
favored only under perturbed FT conditions (i.e. BL injections). Finally, during the three
precipitation events (on $1^{st}$, $4^{th}$ and $7^{th}$ March) we can identify again that the wet removal of
the more hygroscopic aerosol (Fig. 2) suppresses the presence of cloud-activating particles, at
times depleting the atmosphere almost completely from CCN (Fig. 3). This is clearly shown
on the $1^{st}$ and the $7^{th}$ of March, when the CCN number measured at 0.74% supersaturation
drops below 10 $cm^{-3}$, which is extremely low for BL concentrations.

The aerosol hygroscopicity parameter derived from all CCN data collected between 24

of February and 8 of March is presented in Figure 4a. The red solid line represents the hourly
averaged hygroscopicity values over one complete instrument supersaturation cycle. The
hygroscopic properties of the particles at WFJ vary as a function of supersaturation, exhibiting
on average lower values (∼0.1) at high SS and higher values (∼0.3) at the lower SS. Since the
supersaturation inversely depends on particle size, Figure 4a indicates that the hygroscopicity
of the particles drops by almost 60% as the particles are getting smaller (i.e. as the
supersaturation increases). Table 2 summarizes the mean values of $\kappa$ and $D_{cr}$ and their standard
deviations, as calculated for each measured SS. The anticorrelation seen between the instrument
SS and $D_{cr}$ is reasonable, if we consider that the latter represents the minimum activation
diameter in a population of particles, and therefore only the particles with a $D_{cr} > 193.54$ nm
are able to activate into cloud droplets at low SS values (0.09%). The hourly averaged
hygroscopicity at each SS slot falls within a range of ∼0.2 and ∼0.3, which is a well
representative value for continental aerosols (Andreae and Rosenfeld, 2008; Rose et al., 2008).





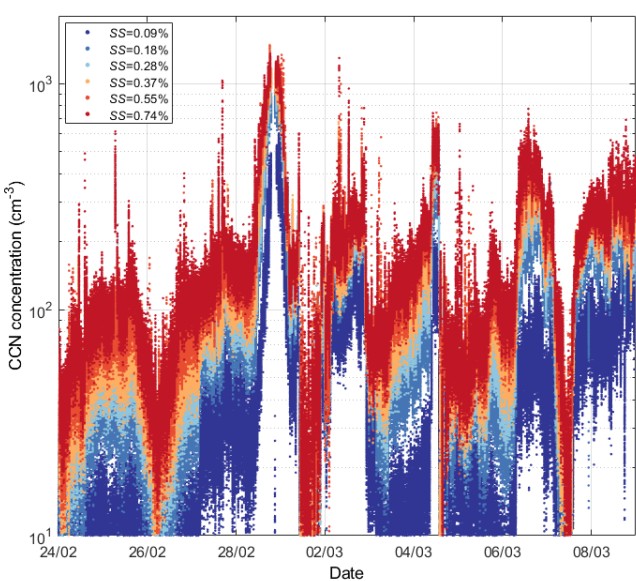


**Figure 3.** Timeseries of CCN number concentrations (cm$^{-3}$) at WFJ for different levels of
supersaturation (*SS*) with respect to water between 24 February and 8 March 2019.


**Table 2.** Average $\kappa$ and $D_{cr}$ values at WFJ for each *SS* measured over the period of interest.

Uncertainty for each value is expressed by the standard deviation.

| SS (%) | $\kappa_{mean}$ | $D_{cr,mean}$ |
|--------|-----------------|---------------|
| 0.09 | 0.26 ± 0.10 | 193.54 ± 29.58 |
| 0.18 | 0.31 ± 0.13 | 116.80 ± 22.20 |
| 0.28 | 0.25 ± 0.13 | 96.69 ± 21.62 |
| 0.37 | 0.24 ± 0.13 | 82.67 ± 20.93 |
| 0.55 | 0.20 ± 0.12 | 68.30 ± 20.95 |
| 0.74 | 0.19 ± 0.11 | 58.11 ± 17.54 |


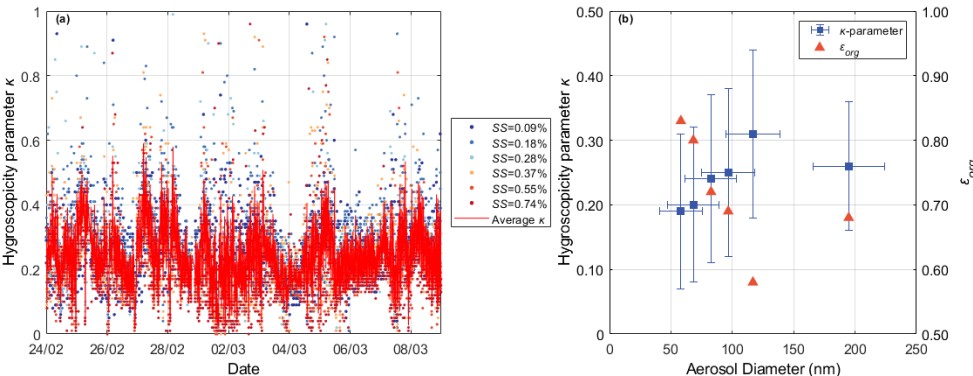


**Figure 4.** (a) Timeseries of the hygroscopicity parameter $\kappa$ at WFJ at different levels of
instrument supersaturation (0.09–0.74%) throughout the period of interest. The solid red line
indicates the hourly averaged $\kappa$ timeseries over a complete $SS$ cycle. (b) Size-resolved aerosol
hygroscopicity (blue squares) and the respective $\varepsilon_{org}$ (orange triangles) calculated for the WFJ
site.


The hygroscopicity parameter along with the inferred $\varepsilon_{org}$ (Eq. 2) is shown in Figure 4b

as a function of particle size. Compared to smaller particles, the higher $\kappa$ of larger particles
(>100 nm) is consistent with them being more aged and with a lower fraction of organics. The
smaller particles are possibly enriched in organic species, which is consistent with the notion
that airmasses in the valley can contain large amounts of freshly emitted BB smoke with lower
$\kappa$. Aerosol particles in the FT are considerably more aged (Seinfeld and Pandis, 2006) and
exhibit higher values of $\kappa$ and consequently lower values of $\varepsilon_{org}$. The chemical composition of
sub-100 nm particulate matter was therefore presumably dominated by organic material
transported from the valley, while the higher $\kappa$ values characterizing the larger particles are
consistent with the more aged character of free tropospheric aerosols (e.g., Jurányi et al., 2011).
The higher $\varepsilon_{org}$ inferred for the smaller particles suggests that mixing between fresh emissions
in the valley and the free tropospheric aerosol might also be taking place at WFJ.

3.2 Potential cloud droplet number concentration and maximum supersaturation
*3.2.1 Method evaluation*
Figure 5 presents an overview of each measurement carried out by the HoloBalloon. Three
cloud events are sampled during the 7[th] and the 8[th] of March, a more detailed description of
which can be found in Ramelli et al. (2020b, c). The observed low-level clouds are likely
produced by orographic lifting when the low-level flow is forced to ascent over the local
topography from Klosters to WOP producing local updrafts and thus water supersaturated



conditions. The potential droplet formation is evaluated using the updraft velocities PDF
calculated for each cloud period (Section 2.3). On March 8, the disdrometer recorded rainfall
over WOP, starting a few minutes after the development of the observed cloud system,
reflected in the gap of updraft velocity timeseries (Fig. 5f). In this case, to determine a relevant
updraft velocity from the wind lidar measurements representative of Cloud 3, we focused on a
15-min time period, between 16:05 and 16:20, before the precipitation occurrence. The
Gaussian fit to the updraft velocity gave a distribution with $\sigma_w = 0.24$ and $0.16$ ms$^{-1}$ for the first
two clouds present on the 7th of March, and, $\sigma_w = 0.37$ ms$^{-1}$ for the cloud system observed on
the 8th of March. The $w^*$ values used to apply the droplet parameterization are thus between
$0.1$-$0.4$ ms$^{-1}$ (Section 2.3).

The cloud LWC measurements from the holographic imager display significant
temporal variability that is also related to variations in the altitude of the tethered balloon
system, as it tends to follow an adiabatic profile (Fig. 5a, b). Deviations from the adiabatic
LWC profile are likely caused by entrainment of dry air within the low-level clouds.
Throughout the two-day dataset presented in Figure 5, the HoloBalloon system samples at
altitudes lower than 300 m AGL, providing observations that are representative of BL
conditions. The observed $N_d$ timeseries collected at WOP are illustrated in Figures 5c and d.
The measurements corresponding to LWC $<0.05$ gm$^{-3}$ are filtered out from the analysis,
assuming that they do not effectively capture in-cloud conditions. A similar criterion for LWC
was also applied in Lloyd et al. (2015) to determine the periods when clouds were present over
the Alpine station of Jungfraujoch. Since the measured cloud properties have finer resolution
(10-20 secs) than the predicted ones, the observed dataset is averaged every 2 minutes. On
March 7, the balloon-borne measurements were taken in a post-frontal air mass (i.e. passage of
a cold front in the morning) and indicated the formation of two low-level liquid layers (Fig. 5c)
over WOP, which is attributed to low-level flow blocking (Ramelli et al., 2020b). During the
first cloud event, an $N_d$ of up to $\smile 100$ cm$^{-3}$ was recorded, while slightly increased $N_d$ in the
range of $\smile 50$-$120$ cm$^{-3}$ is visible during the second cloud event. On March 8, a small-scale
disturbance passed the measurement location Davos, which brought precipitation (Ramelli et
al., 2020c). During the passage of the cloud system, the in-situ measurements collected at WOP
revealed the presence of a persistent low-level feeder cloud confined to the lowest 300 m of
the cloud. The mixed-phase low-level cloud that is shown in Figure 5d, turned into an ice-
dominated low-level cloud after 18 UTC (not shown). Throughout this event, $N_d$ seems to range



between ~100-350 cm$^{-3}$ (Fig. 5d), while the observed ICNC was in the range of ~1-4 L$^{-1}$ (see
Fig. 6b in Ramelli et al., 2020c).

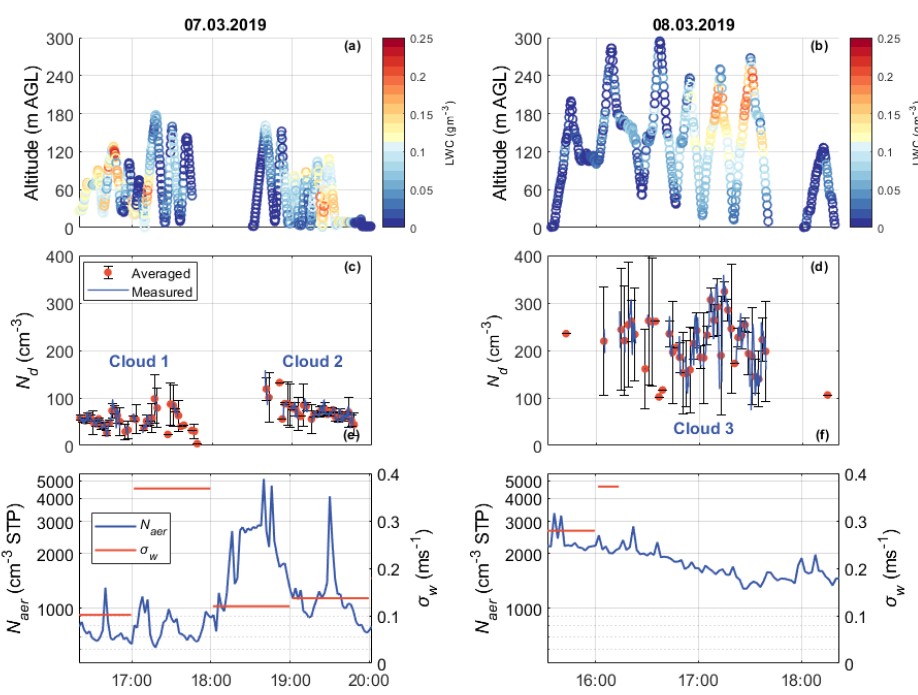


**Figure 5.** Timeseries of the 7$^{th}$ (left panels) and the 8$^{th}$ (right panels) of March, showing the
vertical profiles of the LWC (gm$^{-3}$) in (a) and (b), the filtered (blue lines) and the 2-minute
average (orange dots) $N_d$ (cm$^{-3}$) measured at WOP with the HoloBalloon platform in (c) and
(d), and the corresponding SMPS aerosol concentrations (cm$^{-3}$ STP) (blue line) and the hourly
wind-lidar derived $\sigma_w$ values (ms$^{-1}$) (orange stars) in (e) and (f). Error bars represent the
standard deviation of $N_d$ during the averaging period.

According to Figures 5e and f, low $N_{aer}$ (<10$^3$ cm$^{-3}$) and intermediate $\sigma_w$ values are

representative of the period throughout which the first cloud formed, while up to 4 times higher
$N_{aer}$ is observed during the following two cloud events, with relatively low $\sigma_w$ values
characterizing the second cloud compared to the third one. These contrasted aerosol and
vertical velocity regimes, in which the observed clouds are formed, offer a great opportunity
to test how the proposed methodology performs under a wide range of aerosol and velocity
conditions. Indeed, the mean cloud droplet diameters exhibit a wide range of values, which for
WOP range between 10 μm and 17 μm on March 7, and 8 μm to 12 μm on March 8 (not shown).



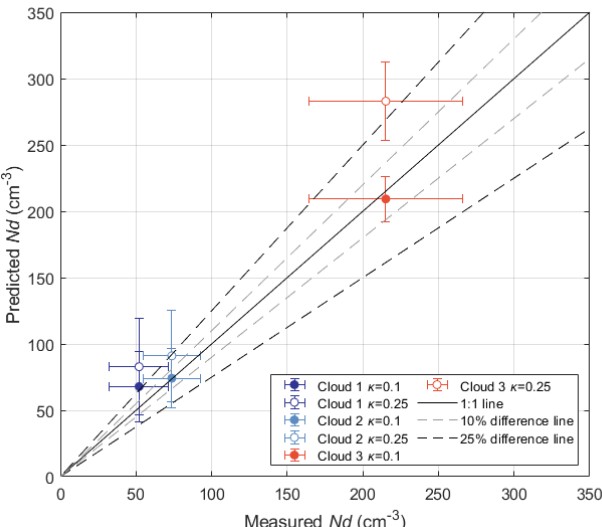


**Figure 6.** Comparison between average predicted $N_d$ (cm$^{-3}$) with the droplet activation parameterization and $N_d$ (cm$^{-3}$) observed during the three cloud events on the 7$^{th}$ (blue and cyan) and the 8$^{th}$ of March (orange) 2019. For all three cloud events droplet closure is performed assuming a $\kappa$ parameter of 0.1 (filled circle) and 0.25 (empty circle). The error bars represent the standard deviation of $N_d$ during each cloud event.


The $N_d$ closure performed for the three cloud events observed over WOP during the last
two days of the period of interest is presented in Figure 6. The parameterization predictions
agree to within 25% with the in-situ cloud droplet number concentrations. A similar degree of
closure is frequently obtained for other in-situ studies (e.g., Meskhidze et al., 2005; Fountoukis
et al., 2007; Morales et al., 2011; Kacarab et al., 2020), which however faced on liquid-phase
clouds. Here we show that the methodology can also work for mixed-phase clouds (i.e. Cloud
3 in Fig. 6). Furthermore, the parameterization predictions indicate that the best fit is achieved
using a $\kappa$ of ⁓0.1. Aerosol concentrations at WOP are likely dominated by lower $\kappa$ values,
indicating that the particles are getting richer in organic material, compared to WFJ, which
supports the aerosol analysis carried out in Section 3.1. These results are robust, indicating that
for non-precipitating BL clouds the proposed calculation method captures cloud droplet
formation at WOP and WFJ.

*3.2.2 Droplet formation at WOP and WFJ*
According to the methodology proposed in Section 2.3, using the measured aerosol number
size distribution, the estimated chemical composition and the observed updraft velocity range,





we determine $N_d$ and $S_{max}$ that would form over both measurement sites. We assume a $\kappa$ of 0.25
to calculate the potential droplets for WFJ according to our CCN-derived hygroscopicity values
(Table 2) and given that $S_{max}$ usually ranges between ∽0.1-0.3%. In estimating the potential
droplets for WOP, we use a $\kappa$ of 0.1 given that aerosol is likely strongly enriched in organics;
the good degree of closure that this value provides supports its selection (Section 3.2.1). Figure
7 depicts the potential $N_d$ and the corresponding $S_{max}$ timeseries calculated for WOP (orange
dots) and WFJ (blue dots) using cloud updraft velocities that are indicative of the $\sigma_w$ range,
being 0.1, 0.3, 0.6 and 0.9 ms$^{-1}$. The same behavior is seen for all four $\sigma_w$ values selected while,
as expected, larger values of $N_d$ and $S_{max}$ are achieved at higher $\sigma_w$. During the first days of the
period of interest, the calculated $N_d$ at WOP (Fig. 7a, c, e, g) is up to 10 times larger than at
WFJ, despite the lower $\kappa$ values characterizing its aerosol population. WFJ tends to have lower
$N_d$ due to the lower $N_{aer}$ recorded. It is also important to highlight the anticorrelation between
$S_{max}$ and $N_d$ values arising from the nonlinear response of droplet number and maximum cloud
parcel supersaturation to fluctuations in the available aerosol/CCN concentrations (Reutter et
al., 2009; Bougiatioti et al., 2016; Kalkavouras et al., 2019). Higher $N_{aer}$ elevates $N_d$ values.
The available condensable water is then shared among more growing droplets, depleting the
supersaturation. Even more interesting is the fact that until February 28 the calculated $N_d$
timeseries at WOP show a pronounced diurnal cycle, similar to the total $N_{aer}$ timeseries (Section
3.1). Lower $N_d$ values are visible during nighttime due to the limited turbulence. Droplet
concentrations at WFJ do not follow a diurnal pattern in contrast to the aerosol data (Fig. 1a).
However, the activation fraction (i.e. $N_d/N_{aer}$) at WFJ displays a clear diurnal variability until
the end of February (supplement Fig. S3).

Through comparison with the MeteoSwiss precipitation measurements at WFJ (Fig. 4),

it should be emphasized again that during the second sub-period of interest the occurrence of
precipitation is followed by a depression in $N_d$ (Fig. 7a, c, e, g) and a concurrent increase in
$S_{max}$ reaching up to ∽1% (Fig. 7b, d, f, h). Especially at WFJ $N_d$ drops almost to zero on the
1st, the 4th and the 7th of March, when precipitation is most intense (orange-shaded areas on
Fig. 7). These trends are related to the washout of hygroscopic material observed at WFJ (Fig.
2) leading to the extremely low CCN concentrations (∽10 cm$^{-3}$) measured during these three
days. During the first two precipitation events, the aerosol concentrations are relatively high,
compared to the third event, with concentrations reaching up to ∽300 cm$^{-3}$ at both stations. The
small activation fraction (supplement Fig. S3) combined with the high $S_{max}$ values indicates
once more that small particles that activate into cloud droplets only above 0.3 to 0.5% of





supersaturation are present at both stations. However, this behavior is not seen on March 7 for
WFJ.

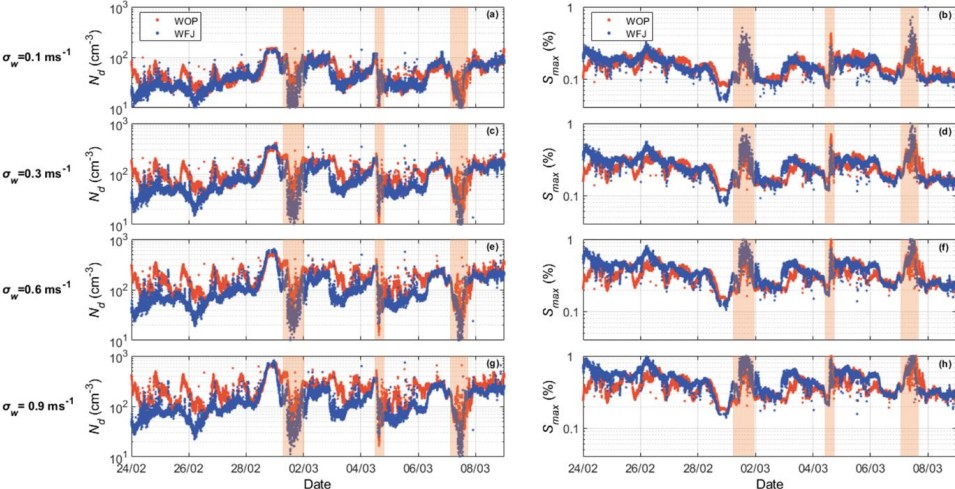


**Figure 7.** Calculated timeseries of $N_d$ (cm$^{-3}$) (left panels) and $S_{max}$ (%) (right panels), for updraft
velocities of $\sigma_w = 0.1$ ms$^{-1}$ in a and b, 0.3 ms$^{-1}$ in c and d, 0.6 ms$^{-1}$ in e and f, and 0.9 ms$^{-1}$ in g
and h, during the period of interest at WOP (orange dots) and WFJ (blue dots). The orange-
shaded areas represent the periods when precipitation is recorded at WFJ site, followed by a
depression in droplet number.


*3.2.3 Droplet behavior under velocity-limited conditions*
Combining the potential $N_d$ and the corresponding $S_{max}$ with the aerosol size distribution data,
it is important to identify regimes where clouds formed are sensitive to vertical velocity
changes or they are sensitive to variations in aerosol concentrations. Figure 8 shows the
response of the calculated $N_d$ to changes in total aerosol concentration for a representative range
of updraft velocities prevailing over WOP (top panels) and WFJ (bottom panels). The data are
colored by the respective $S_{max}$ achieved in cloudy updrafts. For low $\sigma_w$ values (Fig. 8a, d) we
can identify that above an aerosol concentration of ∽300 cm$^{-3}$, the maximum $N_d$ at both stations
reaches a plateau, where its incremental change becomes insensitive to further aerosol changes.
At WFJ, the same behavior is seen for intermediate $\sigma_w$ values and $N_{aer} \gtrsim 1000$ cm$^{-3}$ (Fig. 8f).
The horizontal dashed lines plotted on Figure 8 (a), (e) and (f) illustrate this plateau, which is
termed limiting droplet number ($N_d^{lim}$), following Kacarab et al. (2020). $N_d^{lim}$ is reached owing
to the extreme competition of the high aerosol concentrations for the available condensable
water. In this regime, the clouds are insensitive to aerosol variations and the modulation of the



droplet number is driven mostly by the cloud dynamics, hence the updraft velocity variability.
Consequently, when $N_d$ approaches $N_d^{lim}$ the underlying dynamics control the cloud
microphysics. Within the velocity-limited regime of droplet formation, we can notice that the
corresponding $S_{max}$ values are low (<0.1 %), reflecting the severe water vapor limitation that
allows only a few particles to activate into cloud droplets. Conversely, when $S_{max}$ in clouds
exceeds 0.1% droplet formation in the BL of both measurement sites is always in the aerosol-
limited regime, as the maximum supersaturation is high enough to activate almost all particles
except for the very small ones. In the aerosol-limited regime, $N_d$ never exceeds the
characteristic limit, $N_d^{lim}$. The changeover from aerosol- to velocity-limited conditions also
depends on the change in slope of CCN spectra (Twomey, 1977), and that is why the transition
occurs over a region of Na (Figure 8).

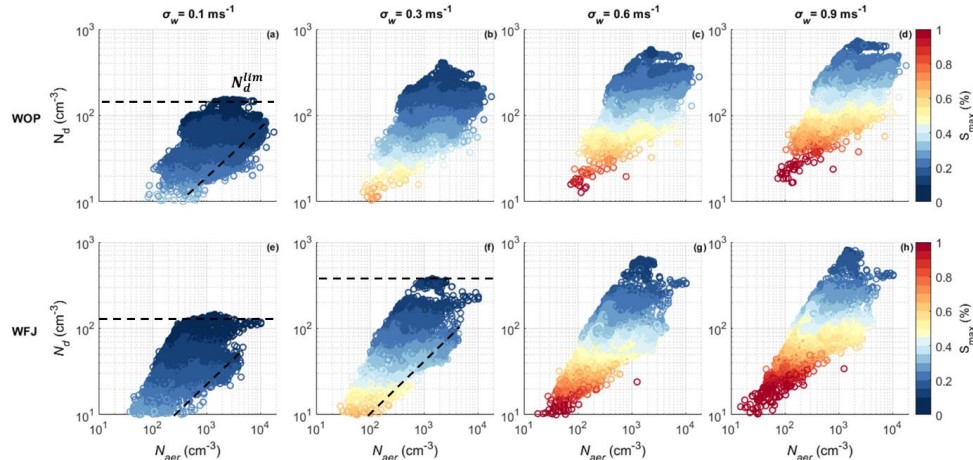


**Figure 8.** $N_d$ (cm$^{-3}$) vs. $N_{aer}$ (cm$^{-3}$), for updraft velocities of $\sigma_w$ = 0.1 ms$^{-1}$ in a and e, 0.3 ms$^{-1}$ in
b and f, 0.6 ms$^{-1}$ in c and g and 0.9 ms$^{-1}$ in d and h, during the period of interest at WOP (top
panels) and WFJ (bottom panels). Data are colored by $S_{max}$ (%).

An alternative way of examining the $N_d^{lim}$ response to changes in the dispersion of

updraft velocity is shown in Figure 9. The limiting droplet number is defined within the
vertical-velocity regime, assuming that this regime prevails when $S_{max}$ drops below 0.1%. At
WOP, droplet formation is in the velocity-limited regime only for low $\sigma_w$ values, namely 0.1
and 0.2 ms$^{-1}$, when the activated particles have more time to deplete the gas phase, and the $S_{max}$
that is reached is that required to activate only the largest particles. At WFJ the prevailing
dynamics create velocity-limited conditions even for more convective boundary layers when



$\sigma_w$ reaches up to 0.5 ms$^{-1}$. $N_d^{lim}$ (cm$^{-3}$) is linearly correlated with $\sigma_w$ (ms$^{-1}$) which can be
described as $N_d^{lim} = 1110.6\,\sigma_w + 24.2$ (Fig. 9). As a result, doubling $\sigma_w$ from 0.1 to 0.2 ms$^{-1}$
increases $N_d^{lim}$ by ⌣60 % for both sites, while transitioning from 0.2 to 0.4 ms$^{-1}$ further
increases $N_d^{lim}$ by ⌣45 %, and finally an additional ⌣20 % increase in $N_d^{lim}$ occurs for WFJ
for the 0.4-0.5 ms$^{-1}$ velocity range. Remarkable agreement is seen for corresponding trends
between $N_d^{lim}$ and $\sigma_w$ calculated for marine Stratocumulus clouds formed under extensive BB
aerosol plumes over the Southeast Atlantic (SEA) Ocean (Kacarab et al., 2020), along with BL
clouds formed in the Southeast United States (SEUS) (Bougiatioti et al., 2020). This realization
is important as it implies that for regions where velocity-limited conditions are expected (i.e.
under particularly high particle loads), $N_d \approx N_d^{lim}$ and the $N_d^{lim}$-$\sigma_w$ relationship can be used to
diagnose $\sigma_w$ from retrievals of droplet number for virtually any type of BL cloud.

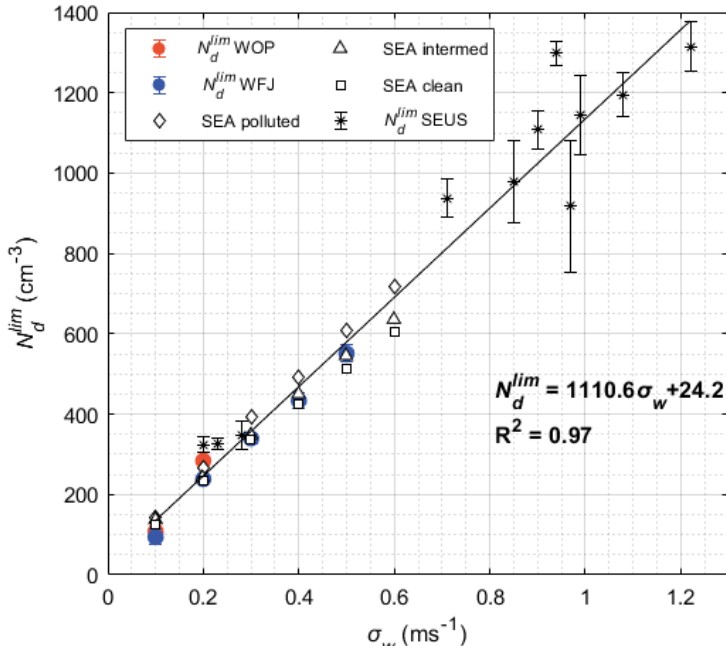


**Figure 9.** Limiting droplet number (cm$^{-3}$) against the standard deviation of updraft velocity
(ms$^{-1}$), calculated when vertical-velocity conditions are met over WOP (orange trace) and WFJ
(blue trace) sites throughout the period of interest. Superimposed are the corresponding values
calculated for polluted (rhombuses), intermediate (triangles) and clean (squares) conditions
over the SEA Ocean and over the SEUS (asterisks).



*3.2.4 $\sigma_w$ and observed $N_d$ determine if droplet formation is aerosol- or velocity-limited*
Observations of $N_d$ when compared against $N_d^{lim}$ can potentially be used to deduce if droplet
formation is velocity- or aerosol-limited. This is important because it indicates whether aerosol
fluctuations are expected to result in substantial droplet number responses in clouds. The strong
correlation between $\sigma_w$ and $N_d^{lim}$ enables this comparison. From the $\sigma_w$ timeseries together with
the linear $N_d^{lim}$-$\sigma_w$ relationship (Section 3.2.3; Fig. 9) we obtain estimates of $N_d^{lim}$ for both
measurement stations (black dashed line in Fig. 10a, b) and the ratio $N_d/N_d^{lim}$ (magenta dotted
lines in Fig. 10a, b). The $N_d$ timeseries calculated for WOP tend to be approximately one third
of $N_d^{lim}$ for most of the observational period (colored circles in Fig. 10a, b), while for WFJ the
same ratio is even lower ⌐1/4. Focusing on the relatively short periods when $S_{max}$ values drop
below 0.1%, we estimate that droplet formation over both measurement sites enters a velocity-
limited regime when the ratio $N_d/N_d^{lim}$ exceeds a critical value of 0.5, with the most prevalent
value being at ⌐0.7 (supplement Fig. S4).

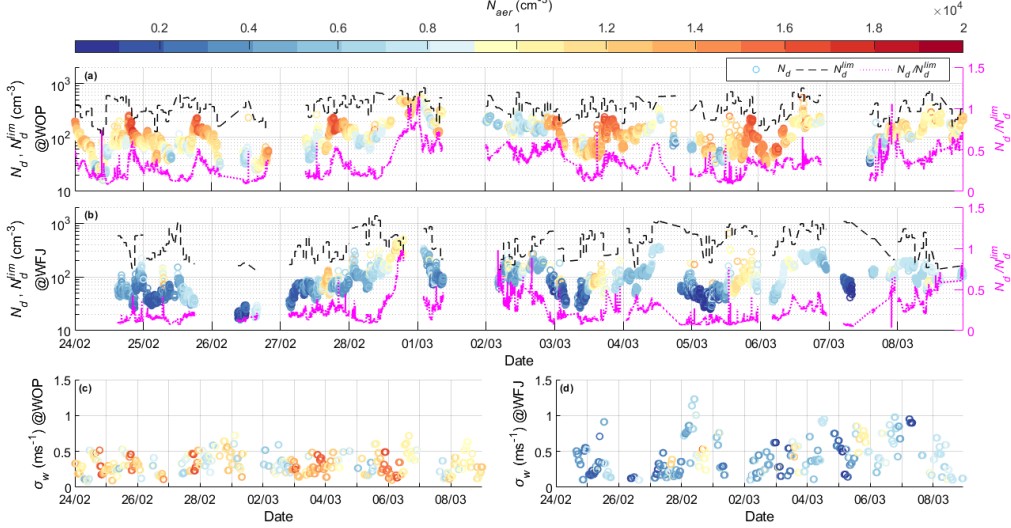


**Figure 10.** Timeseries of potential $N_d$ (cm⁻³) (circles colored by total aerosol number) along
with $N_d^{lim}$ (cm⁻³) (black dashed line) and the ratio between those two (i.e. $N_d/N_d^{lim}$) (magenta
dotted line), together with the timeseries of the calculated standard deviation of updraft
velocities (ms⁻¹), as estimated for WOP (a, c) and WFJ (b, d).

Throughout the period of interest velocity-limited conditions are met at WOP (WFJ)

with a frequency of ⌐0.5% (⌐2.5%) of the total time, reflecting again the sensitivity of droplet
formation to aerosol fluctuations. During nighttime however, when lower $\sigma_w$ values (⌐0.1 ms⁻





654 $^1$) are recorded at WOP (Fig. 10c), we can observe some short periods characterized by

655 intermediate to high aerosol levels (> 1000 cm$^{-3}$) when the ratio $N_d/N_d^{lim}$ exceeds ~0.5,

656 indicating that droplet variability is driven by updraft velocity. The standard deviation of

657 updraft velocities calculated at WFJ do not display a clear temporal pattern (Fig. 10d) but are

658 generally higher than those recorded at the valley site. This is expected considering the

659 steepness of the topography than can cause updraft velocities to be higher, especially for air-

660 masses approaching the site from the north-easterly directions. Over the high mountain-top site

661 cloud formation is in the velocity-limited regime (i.e. $N_d/N_d^{lim}$>0.5) under high aerosol

662 concentrations (~1500 cm$^{-3}$) and higher $\sigma_w$ conditions (~0.8 ms$^{-1}$). These conditions can be

663 created when polluted air-masses from the valley site are vertically transported to WFJ.

665 **4. Summary & Conclusions**

666 The current study focuses on the aerosol-CCN-cloud droplet interplay in Alpine clouds

667 sampled during the RACLETS field campaign over a two-week period of measurements

668 conducted in the valley (WOP), and at the mountain-top station (WFJ). Our main objective was

669 to investigate the drivers of droplet formation in mixed-phase clouds (MPCs) formed in the

670 region and understand in which situations droplet number is sensitive to aerosol perturbations.

671   Overall, lower $N_{aer}$ were systematically recorded at WFJ, indicating that the site is

672 influenced by FT conditions. Deviations from this behavior are observed during fair weather

673 conditions, when injections from the BL of lower altitudes can cause up to an order of

674 magnitude elevation in the aerosol concentrations measured at WFJ. Combining the particle

675 size distribution and CCN number concentration measured at WFJ, the average hygroscopicity

676 parameter $\kappa$ is about 0.25, consistent with expectations for continental aerosol. The size-

677 dependent $\kappa$ reveals that accumulation mode particles are more hygroscopic than the smaller

678 ones, which we attribute to an enrichment in organic material associated with primary

679 emissions in the valley. The hygroscopicity of the particles at WFJ exhibit variations until

680 February 28, which could reflect BL injections from the valley. Precipitation events occurring

681 during the second sub-period of interest, efficiently remove particles, sometimes leaving some

682 less hygroscopic particles.

683   Wind lidar products collected at WOP constrain the PDF of updraft velocity, which

684 combined with observed size distributions and hygroscopicity can be used to calculate the $N_d$

685 in clouds. We show predictions to agree within 25% with the limited observations of droplet

686 number available. While this degree of closure has been achieved in past studies for liquid-



phase clouds, it has not been done at temperatures below freezing and with clouds containing
ice – as done here.

When $\sigma_w$ is equal to 0.1 ms$^{-1}$ droplet formation over both measurement sites is always

aerosol-limited if aerosol concentrations fall below ~300 cm$^{-3}$. For intermediate and higher $\sigma_w$
conditions (>0.3 ms$^{-1}$) the same behavior is seen, but the aerosol-limited regime is extended to
higher aerosol concentrations ~10$^3$ cm$^{-3}$. When droplet formation is within the velocity-limited
regime, it does not exceed a characteristic value, $N_d^{lim}$, that depends on $\sigma_w$. We found that $N_d^{lim}$
is reached when sufficient aerosol is present to decrease $S_{max}$ below 0.1%, and corresponds to
when $N_d/N_d^{lim}$ is above 0.5 for both measurement sites (with a most likely transition value at
0.7). Based on this understanding, we deduce that droplet formation throughout the period of
interest appears most of the time to be aerosol-limited. More specifically, at the valley site,
WOP, clouds become sensitive to updraft velocity variations only during nighttime, when the
BL turbulence is low. Conversely, velocity-limited conditions are encountered at WFJ, during
periods characterized by elevated aerosol and CCN concentration levels (>10$^3$ cm$^{-3}$) and higher
$\sigma_w$ values (~0.8 ms$^{-1}$). Although variations in vertical velocity have not always been found to
be the strongest factor influencing the cloud microphysical characteristics, correct
consideration of updraft velocity fluctuations is crucial to fully understand the drivers of droplet
variability and the role of aerosol as a driver of $N_d$ variability.

Interestingly, we find that the same linear relationship between $N_d^{lim}$ and $\sigma_w$ that

describes the droplet formation during RACLETS holds for warm boundary layer clouds
formed in the SE US (Bougiatioti et al., 2020) and in the SE Atlantic (Kacarab et al., 2019).
This implies that the $N_d^{lim}$-$\sigma_w$ relationship may be universal, given the wide range of cloud
formation conditions it represents. If so, measurements (or remote sensing) of droplet number
and vertical velocity distribution alone may be used to determine if cloud droplet formation is
susceptible to aerosol variations or solely driven by vertical velocity – without any additional
aerosol information.

Approaching velocity-limited conditions also carries important implications for ice-

formation processes in MPCs – as high droplet number means that droplet size and the
probability of riming becomes minimum. Indeed, Lance et al. (2011) saw that the concentration
of large droplets exceeding 30μm diameter – critical for rime splintering or droplet shattering
to occur – drops considerably for polluted Arctic mixed-phase clouds with liquid content ~ 0.2
g m$^{-3}$ and droplet number ~ 300-400 cm$^{-3}$. Assuming that these levels of droplet number reflects
$N_d^{lim}$, the corresponding $\sigma_w$ is 0.3-0.35 ms$^{-1}$ (Fig.9), which is characteristic for Arctic stratus.



The same phenomenon can also occur in the Alpine clouds studied here, given that velocity-
limited conditions ($N_d/N_d^{lim}$>0.5) occurs especially during nighttime (Fig.10). Therefore,
observations of droplet number and vertical velocity distribution (i.e., $N_d^{lim}$) may possibly be
used to determine if SIP from riming and droplet shattering is impeded, and if occurring
frequently enough may help explain the existence of persistent MPCs.

**Data Availability:** The data used in this study can be downloaded from the EnviDat data portal
at https://www.envidat.ch/group/about/raclets-field-campaign. The meteorological
measurements are provided by the Swiss Federal Office of Meteorology and Climatology
MeteoSwiss at https://gate.meteoswiss.ch/idaweb/login.do. The Gaussian fits used for
determining $\sigma_w$ and the droplet parameterization used for the calculations in the study are
available from athanasios.nenes@epfl.ch upon request.

**Author Contributions:** PG and AN designed and initiated the study with methodology and
software developed by AN. The analysis was carried out by PG and AN, with input from ABo,
JW, CM, ZAK, JH, MH, ABe, UL. CCN instrumentation was setup by ABo, aerosol
instrumentation and inlet setup was done by JW, CM and ZAK, cloud data by FR, JH, lidar
data by MH. Instrument maintenance during the field campaign was carried out by JW and
CM. Data curation was provided by PG, AN, JW, CM, FR. The original manuscript was written
by PG and AN with input from all authors. All authors reviewed and commented on the
manuscript.

**Funding:** This study was supported by Swiss Federal state funds, the European Research
Council, CoG-2016 project PyroTRACH (726165) funded by H2020-EU.1.1. – Excellent
Science, and from the European Union Horizon 2020 project FORCeS under grant agreement
No 821205. JW, FR, ZAK, JH, UL acknowledge funding from the Swiss National Science
Foundation (SNSF) grant number 200021_175824.

**Conflicts of Interest:** The authors declare no conflict of interest.



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
