# Peer review of "On the drivers of droplet variability in Alpine mixed-phase 1 clouds 2"

_Atmospheric Chemistry and Physics, 2020_

## Referee Comment (RC1) · Anonymous Referee #1 · 27 Dec 2020

The authors analyze the aerosol size distribution, CCN, hygroscopicity, cloud droplet number concentration, and lidar-derived vertical velocity during the RACLETS field campaign. An adiabatic parcel model is applied to explore the relationship between aerosol property and cloud droplet number concentration in the mixed-phase clouds under various aerosol concentration and vertical velocity. The calculated cloud droplet number concentration is compared with that observed from the tethered balloon system. The aerosol-limited regime and the velocity-limited regime are explored. In general, the paper is well written, and I have some comments listed below that need to be addressed properly before it can be published in Atmospheric Chemistry and Physics.

General comments:

1. Add more discussion about why surface measured aerosol can be used to estimate

cloud droplet number concentration in the cloud above. It might not be true if the orographic clouds are formed somewhere else and drift to the observational site, or the cloud is decoupled with the surface, or aerosol from the long-range transport plays a significant role in cloud droplet activation.

2. There is no in-depth discussion of the properties of the mixed-phase clouds (e.g. INP or ice properties) in the manuscript. Are you sure ALL those clouds are mixed-phase clouds? If not, the title is not accurate. Please consider changing the title and corresponding text in the manuscript or adding evidence to show that those clouds are mixed-phase clouds. In addition, ice processes and collision coalescence process are not considered in the cloud parcel model. The statement "Under such conditions, droplet size tends to be minimal, reducing the likelihood that large drops are present that promote glaciation through rime splintering and droplet shattering" is unfair and misleading in the abstract. Even this statement is true, several crucial steps are needed to prove this statement.

Specific comments:

1. Line 162: "11.5 nm and 469.8 nm" are radius or diameter?

2. Line 280: "... being 200 m and 1100 m AGL for WOP and WFJ". This sentence is not consistent with the previous statement. Only WOP has wind lidar, correct? Why choose 200 m and 1100 m as the altitude of interest?

3. Line 308: Please add pressure and temperature figures to show that "a high-pressure system was dominant over Europe with clear skies and elevated temperatures."

4. Figure 1a: suggest changing the red color of dots for WOP. Red also means high kappa value which is confusing.

5. Line 360 and Figure 2: suggest shading the time period when precipitation occurs in Figure 2, as done in Figure 7. It might be helpful to visualize "a big spike of Naer"

before the third precipitation event.

6. Figure 8: I'm a little bit confused here. So different circles represent calculated Nd for different aerosol size distribution and kappa? The horizontal dashed line represents the limiting droplet number. What about the other dashed line?

———————————————

---

## Referee Comment (RC2) · Jefferson Snider (Referee) · 29 Dec 2020

**Review of "On the drivers of droplet variability in …" by Georgakaki et al., submitted to ACP, November 2020**

**By Jeff Snider, University of Wyoming**

The authors have done a good job of improving their ACPD manuscript.  Below, I provide a detailed review of their ACP submission. In my opinion, the following things should be addressed before the manuscript is published in Atmospheric Chemistry and Physics.

**Most Important:**

In my opinion, it is subjective to bin $\sigma_w$ as 0.1 0.3, 0.6 and 0.9 m/s (Figure 8).  What happens if you re-bin with a different set of $\sigma_w$ values? Please note, I do not feel that a complete answer is on L610 to L614. Given that I consider your setting of $\sigma_w$ to be subjective (Figure 8), and what you say on L610 to L614, it is my opinion that you need to provide evidence (and discussion) of the robustness of the following procedure: Your setting of $\sigma_w$ values, your drawing of horizontal lines (e.g., in Figure 8), and your picking of $N_d^{\lim} - \sigma_w$ pairs for Figure 9.

Related to what I say above, did the papers that resulted from SEA and SEUS examine the robustness of their procedure for determining $N_d^{\lim} - \sigma_w$ pairs? Was their procedure the same as yours?

Here is a related question: Why not draw a horizontal line at the flat-tops (plateaus) of Figures 8d or 8g and conclude that those $N_d^{\lim} - \sigma_w$ pairs should also be included in Figure 9?

Also in my opinion, to refer to the derived $N_d^{\lim}$ (i.e., where the horizontal lines are drawn in Figure 8) as the velocity-limiting concentration requires a caveat. The caveat is that only those aerosol size distributions corresponding to points close to the horizontal line are actually velocity-limited. Consequently, I cannot understand this statement, mostly because it is not clear what you are implying by "regime": "Within the velocity-limited regime of droplet formation, we can notice that the corresponding Smax values are low (<0.1 %),

reflecting the severe water vapor limitation that allows only a few particles to activate into cloud droplets." I'm looking at Figures 8a, 8d, and 8e and in all cases, below where you have drawn the horizontal line, I see data points with Smax > 0.1%. This is particularly evident in Figure 8e. These cases have relatively large $N_d$-to-$N_{aer}$ ratios. Those relatively large ratios concord with those points having relatively a large Smax (i.e., larger than 0.1 %). It follows that those cases *could have* achieved larger $N_d$, for example, had they had the same $N_{aer}$ but fewer large particles (recalling that $N_{aer}$ is controlled by smaller particles), or if they had an Aitken mode at critical supersaturations larger than 0.1 %. In either case, those cases are not velocity-limited, rather, they are aerosol limited. If I have it right, this is your main point. This aerosol limitation is in the sense envisioned by Twomey (1993) where he states that "In a general way, increasing particle numbers must reduce the maximum supersaturation achieved, which means that some previously activated particles may now not be activated. This factor itself tends to reduce droplet numbers somewhat. Hence it is not necessarily true that increasing particle numbers mean a proportional increase in droplet numbers. It is not too difficult to invent distributions of particles such that an increase in their total concentration leads to *reduced* numbers of final droplets". I feel that the authors need to better incorporate what Twomey (1993) was/is getting at. He was explicit! Overall, this paragraph (L577 to L601) needs to be rewritten with improved definition of what you mean by "regime" and with recognition of the shoulders you rest on.

Related to what I said in the previous paragraph, I feel you need to spell out what you mean by velocity-limited and aerosol-limited. Perhaps it's best to do this with a table. As I understand it, velocity-limited implies Nd ~ $N_d^{\lim}$ and Smax < 0.1%. Conversely, aerosol-limited implies Nd < $N_d^{\lim}$ and Smax > 0.1%. Still, I'm puzzled by two things. 1) You use $N_{aer}$ to classify aerosol-limited behavior (L689 to L682) and you also use Smax to classify aerosol-limited behavior (L693 – L696). Which of the second conditions (the Smax second condition or the $N_{aer}$ second condition) is best, in your estimation, to guide future investigators and especially those conducting experiments at locations other than your field site? 2) The table

you can also help those who wonder how to think about a situation with Nd ~ $N_d^{\text{lim}}$, and with Smax > 0.1 %, and what to call that situation.

The issue of who to cite, and who not to cite, is too parochial. Here is one example of this in your manuscript. If you are going to contend that the relationship shown in Figure 9 is significant, for example because it can be used to diagnose $\sigma_w$ from retrievals, then it is important that you reference work that has retrieved (using airborne lidar, for example) droplet concentration. If you don't want to cite the Wyoming team, and want to focus on space borne lidar, then please do so. Perhaps you should cite Danny Rosenfeld's team, and the work they are doing in this arena.

On L721 to L724 you must be careful. The persistence/existence of mixed phase clouds can be controlled by many things other than SIP. For example, the availability of active INPs (need primary ice to get secondary ice) and the temperature regime (Hallett/Mossop occurs over a relatively narrow temperature range). As I pointed out in my pre-review, control of droplet size (and thus SIP) can also come from variation of cloud depth and entrainment, and from processes that broaden the cloud droplet size distribution (updraft variability is one of them). So, I encourage you to more carefully circumscribe what you are saying in lines L721 to L724.

**Less Important, but should also be addressed:**

Figure 8 - What is the purpose of the sloped dashed lines in Figures 8a, 8d, and 8e? Can they be removed? They caused me confusion. Perhaps you could put in the 1 to 10 line…maybe that is what those sloped dashed lines are indicating. If so, tell us that. Finally, better definition (longer and outward directed) of the minor ticks is needed in Figure 8, and I would remove the grid within the panels of Figure 8, its making things murky.

**L474 - What about removal of droplets by riming or by completion for vapor in MP conditions? Did the HOLIMO show evidence of either of those processes in those instances where LWC was depleted relative to adiabatic?**

**About the CFSTGC. My understanding (Snider et al., JGR, 2010) is that two stream-wise temperature gradients are experienced by the particles moving along the flow path. Is this correct for the instrument you operated? If yes, why does your description imply that there is only one stream-wise temperature gradient?**

**L176 – L181 – In addition to the sensitivity of SS to pressure, the measured concentrations are also dependent on pressure via their dependence on aerosol sample flow rate. The latter is reported (by instruments) as a mass or volume flow rate. IMO statements are needed to tell the reader how CCN concentrations were calculated for comparison to the following other concentration measurements: $N_{aer}$, aerosol size distribution, measured cloud droplet number, and theoretical cloud droplet number. One place this is relevant is in Figure 5 where aerosol is reported per cubic centimeter at STP and droplets are expressed per cubic centimeter. There are other examples of this (Figures).**

**L204. Why is this statement relevant to this paper?: "HOLIMO has an open path configuration (i.e. the detection volume lies between the two instrument towers) and thus is also able to measure raindrops up to a size of ∽ 2 mm."**

**L209 to L211. I'm think your statement about the bulk density of liquid is getting in the way of what's important. Everyone knows that the density of liquid is a constant at 1000 kilogram per cubic meter. In my view, a hydrometeor size distribution, measured by HOLIMO, should be sufficient for calculating the (cloud) LWC and (cloud) droplet number concentration, provided the ice particles (typically much larger) can be distinguished from the smaller (cloud) droplets. Please provide discussion on how well this distinction (cloud droplets vs ice) can be made by HOLIMO and what the implication is for estimation of Nd (droplet concentration at D > 6 micrometer) and (cloud) LWC.**

**L210. It's not clear why "measured number concentration" is in this sentence.**

**Table 1.** Droplets smaller than 6 micrometer can, in some instances, contribute significantly to droplet number concentration. Is this discussed?

**L206 to L208.** If I have it correct, the WOP is the lower of the two sites. So, you only have cloud microphysical data (HOLIMO) when the WOP was at AGL heights greater than the cloud base? Is this explained?

**L231.** Is the DBS acronym needed for this manuscript?

**L248.** What's relevant here is the value (constant ?) applied for the surface tension, not the value of the universal gas constant. BTW, you have already defined the density of liquid water (cloud microphysical section).

**L286.** I commented on the w-star approach in a review of the Morales and Nenes (2010). Since you are presenting $\sigma_w$ in Figures 5, 7, 8, 9, and 10 please double check that the transformation from $\sigma_w$ to w-star is applied in the activation calculations.

**L335.** The Seinfeld and Pandis (2006) book is enormous. Please specify where in the text the authors conclude this. It is better to reference journal paper (s) which concludes that in aged air the concentration is lower, the accumulation mode is pronounced, and the hygroscopicity is enhanced.

**L357.** IMO, you should introduce, parenthetically, Figure 2 before going into this discussion of the weather-impacted data.

**L360.** Precipitation rate maximizes at 1.1 mm / hr, not "up to ~ 7 mm / hr".

**L372.** The "…March nucleation processes.." should be described differently. You are speculating about the removal of aerosol particles that activate and then removed by precipitation, or by precipitation removing aerosol through diffusive and impaction processes. The word "nucleation" here will alert some of your readers incorrectly to "aerosol particle nucleation" (aka, NPF).

**L464.** I do not see a "gap" of vertical winds in Figure 5f. I do see that the red data end at ~ 16:15. The labels (e) and (f) are not correctly placed in Figure 5.

L466 to L469.  In Figure 5d, I see $\sigma_w$ values from 0 m/s to 0.36 m/s. In Figure 5f, I see $\sigma_w$ values from 0.25 m/s to 0.47 m/s. You say that the selected values are 0.24 and 0.16 m/s for 7 March and 0.37 m/s for 8 March.  Perhaps this comment is only for the beta version of the ACPD manuscript.  Please check.

Did you discuss why the time resolution of $\sigma_w$ is so poor in Figure 5d and Figure 5f? I read that the time resolution of the lidar is "up to 5 s" (Table 1).  Do you mean that the time resolution is "no better than 5 s"?

Figures 5c and 5d are using blue to indicate three things. Error bars on averages (are these actually variabilities), a line that connects filtered/measured values, and the "three cloud events observed."  This presentation needs to be improved.  Thank you for correcting this. Yet, is there not a better color to connect the data points?

L523. "focused" -> "faced."

L601 "Na" -> "Naer"

---

## Author Comment (AC1) · 9 Mar 2021

**Response to Referee #1**

The authors would like to thank the reviewer for carefully reading the manuscript and providing thoughtful remarks that have improved the manuscript. The replies to the comments are given below. The referee comments are highlighted in red with our responses in black fonts.

**General comments:**

1. Add more discussion about why surface measured aerosol can be used to estimate cloud droplet number concentration in the cloud above. It might not be true if the orographic clouds are formed somewhere else and drift to the observational site, or the cloud is decoupled with the surface, or aerosol from the long-range transport plays a significant role in cloud droplet activation.

Thank you for raising this point! Throughout the observation period, clouds are formed locally at Wolfgang-Pass (WOP), when the low-level flow is forced to ascent due to the local topography (the very good droplet closure supports this). This could not be repeated for Weissfluhjoch (WFJ) owing to a lack of in-situ data, however the airmasses sampled (i.e. those given as input to the parameterization) are often in the free troposphere, so they should contain the same aerosol as the one used to form the clouds. This does not apply under perturbed free tropospheric conditions prevailing at WFJ (i.e. injections from the boundary layer of lower altitudes), which however bring less hygroscopic particles at the mountain top site. This is now discussed in the revised manuscript.

2. There is no in-depth discussion of the properties of the mixed-phase clouds (e.g. INP or ice properties) in the manuscript. Are you sure ALL those clouds are mixed-phase clouds? If not, the title is not accurate. Please consider changing the title and corresponding text in the manuscript or adding evidence to show that those clouds are mixed-phase clouds. In addition, ice processes and collision coalescence process are not considered in the cloud parcel model. The statement "Under such conditions, droplet size tends to be minimal, reducing the likelihood that large drops are present that promote glaciation through rime splintering and droplet shattering" is unfair and misleading in the abstract. Even this statement is true, several crucial steps are needed to prove this statement.

These are all good points. We focus on liquid cloud droplets that can form in these clouds along with their drivers of variability, given the other RACLETS studies that focus on the ice content of the clouds (in the following paragraph). It is well established that for similar liquid water content, more cloud droplets lead to smaller droplets. It is also clear that in such situations, secondary processes like collision-coalescence and riming reduces in probability. To note this possibility and important implications is in our opinion quite appropriate – especially since observational evidence in other studies for clouds with similar dynamical conditions point to such possibilities (e.g., Lance et al., 2011). Even more so, given the discussion on limiting droplet number, $N_d^{lim}$, which is effectively the upper limit in droplet number and hence the maximum extent of droplet size reduction. We will note, however, that we have not proven these aspects but the statement remains one of the main implications of our study.

Regarding the mixed-phase nature of clouds during RACLETS, there are three studies under review (Ramelli et al., 2020a, b; Lauber et al., 2020) providing extensive descriptions of ice properties during specific cases (22 February, 7 March, and 8 March 2019) within mixed-phase clouds (MPCs). To provide further evidence of the presence of MPCs, we have included a figure in the supplemental material (Fig. S4), showing the estimated degree of riming of 39 dendritic crystals collected at WFJ (in the scope of another analysis) during our period of interest. Characteristic images of the rimed ice particles are provided. All dendrites that were photographed are at least lightly rimed (i.e. riming degree = 1), which is direct evidence for the co-existence of droplets and ice in clouds. A relevant discussion is added under Section 3.2.1 in the revised manuscript.

It is important to mention here that despite the mixed-phase nature of some clouds sampled over the valley site, processes like condensation freezing and the removal of cloud droplets through riming and collision-coalescence are not found to disturb the maximum supersaturation and hence the number of activated droplets predicted by the parameterization. The good degree of droplet closure supports this statement.

**Specific comments:**

1. Line 162: "11.5 nm and 469.8 nm" are radius or diameter?

Here we are referring to electrical mobility diameter. We have now clarified this in the revision.

2. Line 280: "…being 200 m and 1100 m AGL for WOP and WFJ". This sentence is not consistent with the previous statement. Only WOP has wind lidar, correct? Why choose 200 m and 1100 m as the altitude of interest?

We use vertical velocity data extracted at 200 m for WOP, as the wind lidar deployed at this station has no values closer to the ground, while the wind values at the altitude of WFJ (which is located approximately 1 km above WOP) are selected as a proxy of the vertical velocity prevailing at the mountain top. This is the best we can do with the observations at hand, and we will make sure this is clearly stated in the text.

3. Line 308: Please add pressure and temperature figures to show that "a high pressure system was dominant over Europe with clear skies and elevated temperatures."

A new figure (Figure S3) is now added in the supplemental material showing the air temperature and pressure measured by the MeteoSwiss station at WFJ. The two periods of interest are indicated on this figure with black arrows.

4. Figure 1a: suggest changing the red color of dots for WOP. Red also means high kappa value which is confusing.

Good point! We adopted a blue-to-yellow colormap throughout the paper to avoid any confusion.

5. Line 360 and Figure 2: suggest shading the time period when precipitation occurs in Figure 2, as done in Figure 7. It might be helpful to visualize "a big spike of $N_{aer}$" before the third precipitation event.

Thanks for this suggestion. The precipitation events depicted in Figure 2 are shaded in the revised manuscript, as per Figure 7.

6. Figure 8: I'm a little bit confused here. So different circles represent calculated $N_d$ for different aerosol size distribution and kappa? The horizontal dashed line represents the limiting droplet number. What about the other dashed line?

We apologize for this confusion. The colormap represents the maximum in-cloud supersaturation (predicted by the parameterization), not the hygroscopicity parameter. The slopped dashed line was used to highlight the upward trend between $N_{aer}$ and $N_d$, observed within the aerosol-limited regime. We agree that these lines are confusing and hence they are now removed from the revised manuscript.

**References:**

Lance, S., Shupe, M. D., Feingold, G., Brock, C. A., Cozic, J., Holloway, J. S., Moore, R. H., Nenes, A., Schwarz, J. P., Spackman, J. R., Froyd, K. D., Murphy, D. M., Brioude, J., Cooper, O. R., Stohl, A. and Burkhart, J. F.: Cloud condensation nuclei as a modulator of ice processes in Arctic mixed-phase clouds, Atmos. Chem. Phys., 11, 8003–8015, doi:10.5194/acp-11-8003-2011, 2011.

Lauber, A., Henneberger, J., Mignani, C., Ramelli, F., Pasquier, J., Wieder, J. and Lohmann, U.: Continuous secondary ice production initiated by updrafts through the melting layer in mountainous regions, 2020.

Ramelli, F., Henneberger, J., David, R. O., Lauber, A., Pasquier, J. T., Wieder, J., Bühl, J., Seifert, P., Engelmann, R., Hervo, M. and Lohmann, U.: Influence of low-level blocking and turbulence on the microphysics of a mixed-phase cloud in an inner-Alpine valley, Atmos. Chem. Phys. Discuss., doi:10.5194/acp-2020-774, in review, 2020a.

Ramelli, F., Henneberger, J., David, R. O., Bühl, J., Radenz, M., Seifert, P., Wieder, J., Lauber, A., Pasquier, J. T., Engelmann, R., Mignani, C., Hervo, M. and Lohmann, U.: Microphysical investigation of the seeder and feeder region of an Alpine mixed-phase cloud, submitted manuscript, 2020b.

---

## Author Comment (AC2) · 9 Mar 2021

**Response to Referee #2**

The authors would like to thank Prof. Snider for carefully reading the manuscript and providing thorough and constructive remarks that have improved the manuscript. The replies to the comments are given below. The referee comments are highlighted in red with our responses in black fonts.

**Most Important:**

1. In my opinion, it is subjective to bin  $\sigma_w$  as 0.1 0.3, 0.6 and 0.9 m/s (Figure 8). What happens if you re-bin with a different set of  $\sigma_w$  values? Please note, I do not feel that a complete answer is on L610 to L614. Given that I consider your setting of  $\sigma_w$  to be subjective (Figure 8), and what you say on L610 to L614, it is my opinion that you need to provide evidence (and discussion) of the robustness of the following procedure: Your setting of  $\sigma_w$  values, your drawing of horizontal lines (e.g., in Figure 8), and your picking of  $N_d^{lim}$ - $\sigma_w$  pairs for Figure 9.

In Figures 7 and 8 we have shown results of  $N_d$  for representative values of  $\sigma_w$  in the observed range of  $\sigma_w$  values derived from wind lidar measurements (shown in Figures 10c and 10d). These observations clearly show that  $\sigma_w$  varies between 0.1 and 1.0 m/s, with the higher values being recorded at the mountain-top site compared to the valley site.

The horizontal lines are plotted in Figure 8 only when velocity-limited conditions are met, and tend to occur when the predicted  $S_{max} < 0.1$  %. The absence of a horizontal line indicates that  $S_{max}$  is systematically higher than 0.1 % and therefore an  $N_d^{lim}$  was not reached. The picking of  $N_d^{lim}$ - $\sigma_w$  pairs, shown in Figure 9, shows that at Wolfgang-Pass (WOP; orange circles)  $N_d^{lim}$  is achieved only for  $\sigma_w \le 0.2$  m/s, while at Weissfluhjoch (WFJ; blue circles)  $N_d^{lim}$  is reached "even for more turbulent boundary layers" with  $\sigma_w \le 0.5$  m/s.

2. Related to what I say above, did the papers that resulted from SEA and SEUS examine the robustness of their procedure for determining  $N_d^{lim}$ - $\sigma_w$  pairs? Was their procedure the same as yours?

Both studies (Kacarab et al., 2020; Bougiatioti et al., 2020) have followed the same probabilistic approach for computing cloud droplet number as the one we followed here. The only difference is that the updraft velocities were obtained from aircraft observations in cloud legs. The robustness of our approach is supported by the good degree of droplet closure achieved. This is also the case for the study of Kacarab et al. (2020), where calculated droplet numbers in non-precipitating boundary layer clouds were found to agree with the observed ones to within 20%.

**3. Here is a related question: Why not draw a horizontal line at the flat-tops (plateaus) of Figures 8d or 8g and conclude that those $N_d^{lim}$ - $\sigma_w$ pairs should also be included in Figure 9?**

As mentioned in Comment 1 above, the horizontal line in Figure 8 is drawn only for those cases when  $S_{max}$  is found to fall below 0.1 %. This indicates which  $N_d^{lim}$ - $\sigma_w$  pairs will be included later

in Figure 9. We should clarify here that  $N_d^{lim}$  is determined by calculating the averaged  $N_d$  achieved whenever  $S_{max} < 0.1$  % for each examined  $\sigma_w$  value.

4. Also in my opinion, to refer to the derived  $N_d^{lim}$  (i.e., where the horizontal lines are drawn in Figure 8) as the velocity-limiting concentration requires a caveat. The caveat is that only those aerosol size distributions corresponding to points close to the horizontal line are actually velocitylimited. Consequently, I cannot understand this statement, mostly because it is not clear what you are implying by "regime": "Within the velocity-limited regime of droplet formation, we can notice that the corresponding Smax values are low (<0.1 %), reflecting the severe water vapor limitation that allows only a few particles to activate into cloud droplets." I'm looking at Figures 8a, 8d, and 8e and in all cases, below where you have drawn the horizontal line, I see data points with Smax > 0.1%. This is particularly evident in Figure 8e. These cases have relatively large  $N_d$ -to- $N_{aer}$  ratios. Those relatively large ratios concord with those points having relatively a large  $S_{max}$  (i.e., larger than 0.1 %). It follows that those cases could have achieved larger  $N_d$ , for example, had they had the same  $N_{aer}$  but fewer large particles (recalling that  $N_{aer}$  is controlled by smaller particles), or if they had an Aitken mode at critical supersaturations larger than 0.1 %. In either case, those cases are not velocity-limited, rather, they are aerosol limited. If I have it right, this is your main point. This aerosol limitation is in the sense envisioned by Twomey (1993) where he states that "In a general way, increasing particle numbers must reduce the maximum supersaturation achieved, which means that some previously activated particles may now not be activated. This factor itself tends to reduce droplet numbers somewhat. Hence it is not necessarily true that increasing particle numbers mean a proportional increase in droplet numbers. It is not too difficult to invent distributions of particles such that an increase in their total concentration leads to reduced numbers of final droplets". I feel that the authors need to better incorporate what Twomey (1993) was/is getting at. He was explicit! Overall, this paragraph (L577 to L601) needs to be rewritten with improved definition of what you mean by "regime" and with recognition of the shoulders you rest on.

Indeed, Twomey (1993) and others before and after (e.g., Jensen and Charlson, 1984; Ghan et al., 1998, Nenes et al., 2001 and Reutter et al., 2009) recognized the role of water vapor competition on droplet formation. Twomey (1993) discusses this conceptually and states that competition may be fierce enough to reduce droplet number with increasing aerosol (which was later demonstrated by Ghan et al. 1998 to occur for mixtures of sulfate aerosol and sea spray). Reutter et al. (2009) did not focus on such extreme conditions of water vapor competition, but rather situations that are consistent with dominance of anthropogenic pollution in clouds. Indeed, for high enough aerosol amount, droplets in clouds become insensitive to aerosol perturbations, giving rise to the so-called "velocity limited cloud formation". Formally this condition can be expressed in terms of the partial derivative of droplet number to aerosol- the smaller the derivative, the more clouds are velocitylimited (e.g., Morales-Betancourt and Nenes, 2014). Kacarab et al. (2020) used such derivatives to express when clouds respond weakly to aerosol perturbations, i.e., became velocity-limited, for a wide range of ambient size distributions. The derivative became small (i.e., the clouds became velocity-limited) when supersaturation dropped below 0.1% because of the increasingly fierce competition for water vapor during droplet formation. Bougiatioti et al. (2020) also found similar results. We build upon these findings and apply the above to the Alpine aerosols and clouds sampled during the study. We will make these discussions very clear.

5. Related to what I said in the previous paragraph, I feel you need to spell out what you mean by velocity-limited and aerosol-limited. Perhaps it's best to do this with a table. As I understand it, velocity-limited implies  $N_d \sim N_d^{lim}$  and  $S_{max} < 0.1\%$ . Conversely, aerosol-limited implies  $N_d < N_d^{lim}$  and  $S_{max} < 0.1\%$ . Conversely, aerosol-limited implies  $N_d < N_d^{lim}$  and  $S_{max} > 0.1\%$ . Still, I'm puzzled by two things. 1) You use  $N_{aer}$  to classify aerosol-limited behavior (L689 to L682) and you also use  $S_{max}$  to classify aerosol-limited behavior (L693 – L696). Which of the second conditions (the  $S_{max}$  second condition or the  $N_{aer}$  second condition) is best, in your estimation, to guide future investigators and especially those conducting experiments at locations other than your field site? 2) The table you can also help those who wonder how to think about a situation with  $N_d \sim N_d^{lim}$ , and with  $S_{max} > 0.1\%$ , and what to call that situation.

This is a good point.  $N_{aer}$  alone is a weak constraint and it can be used only as a rough indicator for aerosol- or velocity-limited conditions. The primary condition for velocity-limited clouds is  $S_{max} < 0.1$  % and  $N_d \sim N_d^{lim}$ , as supported by the histogram presented in the supplementary material (Fig. S6). We will make these points very clear in the text.

6.The issue of who to cite, and who not to cite, is too parochial. Here is one example of this in your manuscript. If you are going to contend that the relationship shown in Figure 9 is significant, for example because it can be used to diagnose  $\sigma_w$  from retrievals, then it is important that you reference work that has retrieved (using airborne lidar, for example) droplet concentration. If you don't want to cite the Wyoming team, and want to focus on space borne lidar, then please do so. Perhaps you should cite Danny Rosenfeld's team, and the work they are doing in this arena.

When this conclusion was written, we cited the work that was used in assembling the specific figure. The suggestion to refer to studies that adopt airborne (e.g. Snider et al., 2017) or satellite (e.g. the review paper of Grosvenor et al., 2018) cloud droplet number retrievals is also useful. These two reference studies are now added in the revised manuscript.

7. On L721 to L724 you must be careful. The persistence/existence of mixed phase clouds can be controlled by many things other than SIP. For example, the availability of active INPs (need primary ice to get secondary ice) and the temperature regime (Hallett/Mossop occurs over a relatively narrow temperature range). As I pointed out in my pre-review, control of droplet size (and thus SIP) can also come from variation of cloud depth and entrainment, and from processes that broaden the cloud droplet size distribution (updraft variability is one of them). So, I encourage you to more carefully circumscribe what you are saying in lines L721 to L724.

We do not disagree with these comments. However, when we refer to aerosol effects on clouds and the size distribution (which are separate from cloud depth, entrainment and vertical velocity variations), approaching the conditions of  $N_d^{lim}$  is consistent with the maximum reduction in droplet size (even if the distribution broadens). The latter tends to be consistent with reduced riming – compared to if the cloud, everything else identical, has less aerosol present.

**Less important, but should also be addressed:**

1. Figure 8 - What is the purpose of the sloped dashed lines in Figures 8a, 8d, and 8e? Can they be removed? They caused me confusion. Perhaps you could put in the 1 to 10 line...maybe that is what those sloped dashed lines are indicating. If so, tell us that. Finally, better definition (longer and outward directed) of the minor ticks is needed in Figure 8, and I would remove the grid within the panels of Figure 8, its making things murky.

Thank you for the suggestions. The sloped dashed lines are removed from Figures 8a, 8d and 8e, to avoid any confusion. Only the horizontal dashed lines can be found in these figures, indicating the limiting droplet number (if it is reached). The representation of the minor ticks is also improved while the grids are removed.

**2. L474 - What about removal of droplets by riming or by completion for vapor in MP conditions? Did the HOLIMO show evidence of either of those processes in those instances where LWC was depleted relative to adiabatic?**

Thank you for this point. Indeed, within the seeder-feeder situation observed on March 8, HOLIMO recorded a large fraction of rimed particles and graupel, indicating that ice particles gained mass by riming and depositional growth while falling through the mixed-phase cloud layer (Ramelli et al., 2020). This comment is now added in the revised manuscript.

3. About the CFSTGC. My understanding (Snider et al., JGR, 2010) is that two stream-wise temperature gradients are experienced by the particles moving along the flow path. Is this correct for the instrument you operated? If yes, why does your description imply that there is only one stream-wise temperature gradient?

Thank you for bringing up this point. The description is based on the fundamental operation principle of the instrument. Calibration is done based on the delta T across the full column so the resulting supersaturation is calibrated against that temperature difference.

4. L176 – L181 – In addition to the sensitivity of SS to pressure, the measured concentrations are also dependent on pressure via their dependence on aerosol sample flow rate. The latter is reported (by instruments) as a mass or volume flow rate. IMO statements are needed to tell the reader how CCN concentrations were calculated for comparison to the following other concentration measurements:  $N_{aer}$ , aerosol size distribution, measured cloud droplet number, and theoretical cloud droplet number. One place this is relevant is in Figure 5 where aerosol is reported per cubic centimeter at STP and droplets are expressed per cubic centimeter. There are other examples of this (Figures).

Thank you for this comment.  $N_{aer}$  reported in Figure 5 is now converted from STP to ambient conditions, to be consistent with the measurements shown in this figure. In Figures 1a and 2, the aerosol concentrations are reported at standard conditions since our aim was to directly compare two stations located at different altitudes – WOP and WFJ. CCN data were collected only at WFJ

and are expressed at ambient conditions in Figure 3. To determine the predicted cloud droplet number and perform the droplet closure study we used solely in-situ measurements. This is the reason why we have decided to report all concentrations per cubic centimeter throughout the manuscript, except  $N_{aer}$  in Figures 1a and 2. Statements are added in the text to clarify whether the measurements are expressed at ambient or standard conditions.

5. L204. Why is this statement relevant to this paper?: "HOLIMO has an open path configuration (i.e. the detection volume lies between the two instrument towers) and thus is also able to measure raindrops up to a size of  $\sim 2$  mm."

Thank you. This statement is removed.

6. L209 to L211. I'm think your statement about the bulk density of liquid is getting in the way of what's important. Everyone knows that the density of liquid is a constant at 1000 kilogram per cubic meter. In my view, a hydrometeor size distribution, measured by HOLIMO, should be sufficient for calculating the (cloud) LWC and (cloud) droplet number concentration, provided the ice particles (typically much larger) can be distinguished from the smaller (cloud) droplets. Please provide discussion on how well this distinction (cloud droplets vs ice) can be made by HOLIMO and what the implication is for estimation of Nd (droplet concentration at D > 6 micrometer) and (cloud) LWC.

The distinction between cloud droplets and ice crystals is done for particles larger than 25  $\mu$ m diameter based on the particle shape (circular vs non-circular). For particles smaller than 25  $\mu$ m it is challenging to differentiate between the ice and liquid phase owing to resolution limitations of HOLIMO (around 8 times the effective pixel size (3  $\mu$ m) is required to differentiate between liquid and ice). As noted, droplets below 6  $\mu$ m are possible, and if present would result in HOLIMO underestimating their total concentration. The influence of small cloud droplets on the reported LWC is however much smaller, as the contribution of the larger cloud droplets dominates the relevant distribution. This is now mentioned in the text.

7. L210. It's not clear why "measured number concentration" is in this sentence.

Thank you for noting this! This is now removed.

8. Table 1. Droplets smaller than 6 micrometer can, in some instances, contribute significantly to droplet number concentration. Is this discussed?

Good point, a discussion is added in the revised manuscript.

9. L206 to L208. If I have it correct, the WOP is the lower of the two sites. So, you only have cloud microphysical data (HOLIMO) when the WOP was at AGL heights greater than the cloud base? Is this explained?

Indeed, we have cloud microphysical data when the Holoballoon was at AGL heights greater than the cloud base.

**10. L231. Is the DBS acronym needed for this manuscript?**

Good point. It is now removed.

11. L248. What's relevant here is the value (constant?) applied for the surface tension, not the value of the universal gas constant. BTW, you have already defined the density of liquid water (cloud microphysical section).

Thank you for this comment. Here the surface tension of pure water is assumed and this is calculated as a function of temperature.

12. L286. I commented on the w-star approach in a review of the Morales and Nenes (2010). Since you are presenting  $\sigma_w$  in Figures 5, 7, 8, 9, and 10 please double check that the transformation from  $\sigma_w$  to w-star is applied in the activation calculations.

The transformation from  $\sigma_w$  to w\* is actually carried out within the parameterization. The calculated  $\sigma_w$  values are among the inputs required to call the parameterization and this is the reason why we have decided to present  $\sigma_w$  instead of w\* in the figures throughout the manuscript.

13. L335. The Seinfeld and Pandis (2006) book is enormous. Please specify where in the text the authors conclude this. It is better to reference journal paper (s) which concludes that in aged air the concentration is lower, the accumulation mode is pronounced, and the hygroscopicity is enhanced.

The pages referring to the free tropospheric aerosols in Seinfeld and Pandis (2006) are specified in the revised manuscript (pp. 376-378). Here we have also cited some previous studies focused on the mountain-top site of Jungfraujoch (JFJ) in the Swiss Alps. For instance, when free tropospheric conditions prevail at JFJ, Baltensperger et al. (1997) provided evidence that sufficiently aged aerosol is usually found, while during PBL injections Kammermann et al. (2010) and Jurányi et al. (2011) reported an increase of aerosol loadings accompanied by a decrease in aerosol hygroscopicity.

14. L357. IMO, you should introduce, parenthetically, Figure 2 before going into this discussion of the weather-impacted data.

Done!

15. L360. Precipitation rate maximizes at 1.1 mm / hr, not "up to  $\sim$  7 mm / hr".

Thank you, corrected. There was a mistake in Figure 2.

16. L372. The "...March nucleation processes.." should be described differently. You are speculating about the removal of aerosol particles that activate and then removed by precipitation, or by precipitation removing aerosol through diffusive and impaction processes. The word "nucleation" here will alert some of your readers incorrectly to "aerosol particle nucleation" (aka, NPF).

Thanks for this comment. Suggested changes are made.

17. L464. I do not see a "gap" of vertical winds in Figure 5f. I do see that the red data end at  $\sim$  16:15. The labels (e) and (f) are not correctly placed in Figure 5.

Thank you for these comments. All these issues are now addressed in the revised manuscript.

18. L466 to L469. In Figure 5d, I see  $\sigma_w$  values from 0 m/s to 0.36 m/s. In Figure 5f, I see  $\sigma_w$  values from 0.25 m/s to 0.47 m/s. You say that the selected values are 0.24 and 0.16 m/s for 7 March and 0.37 m/s for 8 March. Perhaps this comment is only for the beta version of the ACPD manuscript. Please check.

Did you discuss why the time resolution of  $\sigma_w$  is so poor in Figure 5d and Figure 5f? I read that the time resolution of the lidar is "up to 5 s" (Table 1). Do you mean that the time resolution is "no better than 5 s"?

The temporal resolution of instantaneous velocity from the wind lidar stated in Table 1 is now changed to "5 s max". However,  $\sigma_w$  is determined with much lower frequency. To calculate the  $\sigma_w$  values that are shown in Figures 5e and 5f, the high-resolution wind lidar data was grouped by hour and each fitted to half-Gaussian PDFs with zero mean and standard deviation  $\sigma_w$  (supplement Fig. S2). The  $\sigma_w$  values shown in Figures 5e, 5f and Figures 10c, 10d are calculated per hour.

To determine the  $\sigma_w$  values for the closure study we isolated the segments that correspond to the three cloud events shown in Figures 5c and 5d, and fit to half-Gaussian PDFs, being  $\sigma_w = 0.24$  and 0.16 m/s for 7 March, and, 0.37 m/s for 8 March.

All the above  $\sigma_w$  values are then used to run the droplet number calculation – by converting each to a characteristic velocity (w\*=0.79 $\sigma_w$ ), which is then used by the parameterization to compute droplet number – which is equal to the PDF-averaged droplet number (Morales and Nenes, 2010).

Changes have been made in the Sections 2.3 and 3.2.1 to clarify the above-mentioned points.

19. Figures 5c and 5d are using blue to indicate three things. Error bars on averages (are these actually variabilities), a line that connects filtered/measured values, and the "three cloud events

observed." This presentation needs to be improved. Thank you for correcting this. Yet, is there not a better color to connect the data points?

The error bars represent the standard deviation of the measured droplet numbers (hence the variabilities), not the average deviation (or the mean absolute deviation). Figure 5 is modified in the revised manuscript. The same color (cyan) is now used to represent the 2-minute averaged  $N_d$  and the corresponding error bars, while a black line is used to connect the data points.

20. L523. "focused" -> "faced."

Amended.

**21. L601 "Na" -> "Naer"**

Thank you, corrected.

**References:**

[revised manuscript text omitted]

---

## Referee Report (RR1)

By Jeff Snider, University of Wyoming

This is my third review of the manuscript. In my opinion, the following things should be addressed before the manuscript is published in Atmospheric Chemistry and Physics.

…………………………………………………………………

L546 - L548. "The Gaussian fit to the updraft velocities gave a distribution with σw = 0.24 and 0.16 ms-1 for the first two clouds present on the 7th of March, and, σw = 0.37 ms-1 for the cloud system observed on the 8th of March."

Two comments:

1) The $\sigma_w$ data (these are 1-hour averages) plotted in Fig. 5e indicate a factor of four spread over the Cloud-1 interval. Consequently, the $\sigma_w$ evaluated for Cloud-1 (0.24 m/s; see L546 - L548) does not seem reasonable for either the first part of Cloud-1 (time < 17:00) or for the second part of Cloud-1 (time > 17:00).

2) It's not clear how to reconcile the $\sigma_w$ averages in Fig. 5e with the $\sigma_w$ evaluated for Cloud-2 (0.16 m/s; see L546 - L548). By eye (Fig. 5e), the value for Cloud-2 is $\sigma_w$ ~ 0.12 m/s. It must be that there is there more updraft variability during the times you have cloud data (Fig. 5c, Cloud-2).

Since the Nd closure (Fig. 6) is central to your paper, I think it is appropriate to explore further this aspect of your analysis. Here is my recommendation. Please present averages of $\sigma_w$ (in and updated Figs. 5e and 5f, or in a response) for intervals shorter than 1 hour. For example, present a 10-minute average corresponding to 120 updraft samples (temporal resolution 5 s max). In my opinion, this would make clear the basis for the $\sigma_w$ you report in L546 - L548. It could also make it simpler for you. For example, should you care to rationalize splitting Cloud-1 into an earlier (time < 17:00) and later interval (time > 17:00) interval. Or, it could make it easier for you to argue that the $\sigma_w$ average for time < 17:00 (7 March) is biased low by updraft measurements collected prior to start of Fig. 5e at ~ 16:30.
……………………………………………………………

L556 – L563 "The good agreement between measurements and predictions - even under mixed-phase conditions, reveals that processes like condensation freezing and.."

It's not clear what you are getting at with "condensation freezing." It's established, by your group and others (Korolev et al. 2017), that pathways generating ice beyond a few tens per liter, within moderate updraft (≤ 1 m/s), significantly reduce the Smax. I think this is what Sud et al. (2013) and Barahona et al. (2014) were getting at. Can you rewrite L556 – L563 for clarity?
……………………………………………………………….

……………………………………………………………………………………….

I recommend the following addition of "**by**":

Reutter et al. (2009) pointed out that droplet formation in clouds can be limited by the amount of CCN present (called the "aerosol-limited" regime), or **by** the vertical velocity that generates supersaturation in the cloudy updrafts (called the "velocity-limited" regime).

……………………………………………………………………………………….

"..may decrease CCN activity through the formation of glassy aerosol, has not been assessed in a closure study to date." Is this speculative or is a reference missing?

………………………………………………………………………………

…………………………………………………………………………

 "With box model simulations, Hammer et al. (2015).." These simulations applied a closed adiabatic parcel model, I think. "Box" seems like a rigid container.

……………………………………………………………………….

L299 – L303 "Aiming to examine how $N_d$ responds to different vertical velocity-aerosol situations, as a sensitivity test, potential $N_d$ for both sites are calculated at 10 values of $\sigma_w$ between 0.1 and 1.0 ms-1 that cover the observed range (Section 3.2.4). Note that we use the term "potential" droplet number throughout this study, as its calculation is performed regardless of the actual existence of clouds over the measurement sites."

This application of the word "potential" is useful. Given what you are saying, Section 3.2.1 is _not_ about potential droplet number, rather it's about measured Nd and measured $\sigma_w$ in (near) actual clouds. In contrast, Section 3.2.2 is about potential droplet number.

Here is what I'm advocating for: Please improve the section titles so that they apply your definition (L299 – L303) and especially so for titles of Sections 3.2, 3.2.1, and 3.2.2.

Related to this is L541 – L542:
"Note that the **potential** droplet formation is evaluated using the updraft velocity PDF calculated for each cloud period, rather than the hourly $\sigma_w$ data shown in Figures 5e and 5f (Section 2.3)."

In L541 – L542, it's my opinion, you should remove the word "potential."

…………………………………………………………………………………..

L606 "Lower $N_d$ values are visible during nighttime due to the limited turbulence."

Turbulence is lower near a surface at night, however, turbulence is being prescribed in Fig. 7. The diurnal cycle is explained on L320 – L321: *"$N_{aer}$ at WOP peaks in the evening, reaching up to ∽10⁴ cm-3 presumably because of BB emissions in the valley which seem to stop around midnight (Fig. 1a)."*

……………………………………………………………………………….

…………………………………………………………..

Why did the locations of the SEA points change moderately from Figure 9 (acp-2020-1036-manuscript-version3.pdf) compared to an earlier draft of Figure 9?

…………………………………………………………..

Reference

Korolev, A., McFarquhar, G., Field, P. R., Franklin, C., Lawson, P., Wang, Z., Williams, E., Abel, S. J., Axisa, D., Borrmann, S., Crosier, J., Fugal, J., Krämer, M., Lohmann, U., Schlenczek, O., Schnaiter, M., & Wendisch, M. (2017). Mixed-Phase Clouds: Progress and Challenges, Meteorological Monographs, 58, 5.1-5.50. Retrieved Apr 8, 2021, from https://journals.ametsoc.org/view/journals/amsm/58/1/amsmonographs-d-17-0001.1.xml

---

## Author Response (AR2)

**Response to Referee #2**

The authors would like to thank Prof. Snider for carefully reading the revised manuscript and once more providing very thorough and constructive remarks. Reviewer comments are provided in blue font with our responses in black fonts.

1. L546 - L548. "The Gaussian fit to the updraft velocities gave a distribution with $\sigma_w = 0.24$ and $0.16$ ms$^{-1}$ for the first two clouds present on the 7th of March, and, $\sigma_w = 0.37$ ms$^{-1}$ for the cloud system observed on the 8th of March."

   Two comments:

   1) The $\sigma_w$ data (these are 1-hour averages) plotted in Fig. 5e indicate a factor of four spread over the Cloud-1 interval. Consequently, the $\sigma_w$ evaluated for Cloud-1 (0.24 m/s; see L546 - L548) does not seem reasonable for either the first part of Cloud-1 (time < 17:00) or for the second part of Cloud-1 (time > 17:00).

   2) It's not clear how to reconcile the $\sigma_w$ averages in Fig. 5e with the $\sigma_w$ evaluated for Cloud-2 (0.16 m/s; see L546 - L548). By eye (Fig. 5e), the value for Cloud-2 is $\sigma_w \sim 0.12$ m/s. It must be that there is there more updraft variability during the times you have cloud data (Fig. 5c, Cloud-2).

   Since the $N_d$ closure (Fig. 6) is central to your paper, I think it is appropriate to explore further this aspect of your analysis. Here is my recommendation. Please present averages of $\sigma_w$ (in and updated Figs. 5e and 5f, or in a response) for intervals shorter than 1 hour. For example, present a 10-minute average corresponding to 120 updraft samples (temporal resolution 5 s max). In my opinion, this would make clear the basis for the $\sigma_w$ you report in L546 - L548. It could also make it simpler for you. For example, should you care to rationalize splitting Cloud-1 into an earlier (time < 17:00) and later interval (time > 17:00) interval. Or, it could make it easier for you to argue that the $\sigma_w$ average for time < 17:00 (7 March) is biased low by updraft measurements collected prior to start of Fig. 5e at ~ 16:30.

   Thank you for raising this point. The temporal resolution of the wind lidar products is variable, with the maximum resolution being 5 s (as mentioned in the manuscript). During the two cloud events on 7 March the maximum temporal resolution of the wind lidar was ~ 30 s, allowing us to use ~ 10-20 vertical velocity samples (updrafts + downdrafts) for the suggested 10-min calculations. Given that we fit only the updraft velocities to the half-Gaussian PDFs, which is just a subset of these 10-20 samples, the $\sigma_w$ calculated from the 10-min interval PDF is just too uncertain to be useful. Using the hourly $\sigma_w$ resolves this problem.

   During Cloud-1, the in-cloud updraft variability is indeed high, with higher $\sigma_w$ values recorded after 17:00. A discussion is now added in the revised manuscript mentioning that the $\sigma_w$ derived for Cloud-1 might be biased low by the lower $\sigma_w$ values measured before 17:00. *Nevertheless, the updraft averaging used in the droplet closure study corresponds to the $N_d$ averaging timeperiod and, therefore, we do not expect the degree of closure to be affected.*

2. L556 – L563 "The good agreement between measurements and predictions - even under mixed-phase conditions, reveals that processes like condensation freezing and.."
It's not clear what you are getting at with "condensation freezing." It's established, by your group and others (Korolev et al. 2017), that pathways generating ice beyond a few tens per liter, within moderate updraft ($\leq$ 1 m/s), significantly reduce the Smax. I think this is what Sud et al. (2013) and Barahona et al. (2014) were getting at. Can you rewrite L556 – L563 for clarity?

The sentence has been rephrased.

3. I recommend the following addition of "**by**":
Reutter et al. (2009) pointed out that droplet formation in clouds can be limited by the amount of CCN present (called the "aerosol-limited" regime), or **by** the vertical velocity that generates supersaturation in the cloudy updrafts (called the "velocity-limited" regime).

Thank you, corrected.

4. "..may decrease CCN activity through the formation of glassy aerosol, has not been assessed in a closure study to date." Is this speculative or is a reference missing?

We cannot provide a reference here, since to our knowledge there are no in-situ studies assessing cloud droplet closure in mixed-phase clouds. The sentence is now modified in the revised manuscript, to make this point clearer.

5. "With box model simulations, Hammer et al. (2015).." These simulations applied a closed adiabatic parcel model, I think. "Box" seems like a rigid container.

Thank you for this comment. "Box model" is a term frequently used for 1D models, like what we use here. Nevertheless, we switched to "cloud parcel model" in the revised manuscript.

6. L299 – L303 "Aiming to examine how $N_d$ responds to different vertical velocity-aerosol situations, as a sensitivity test, potential $N_d$ for both sites are calculated at 10 values of $\sigma_w$ between 0.1 and 1.0 $ms^{-1}$ that cover the observed range (Section 3.2.4). Note that we use the term "potential" droplet number throughout this study, as its calculation is performed regardless of the actual existence of clouds over the measurement sites."
This application of the word "potential" is useful. Given what you are saying, Section 3.2.1 is _not_ about potential droplet number, rather it's about measured Nd and measured $\sigma_w$ in (near) actual clouds. In contrast, Section 3.2.2 is about potential droplet number.
Here is what I'm advocating for: Please improve the section titles so that they apply your definition (L299 – L303) and especially so for titles of Sections 3.2, 3.2.1, and 3.2.2.
Related to this is L541 – L542:
"Note that the *potential* droplet formation is evaluated using the updraft velocity PDF calculated for each cloud period, rather than the hourly $\sigma_w$ data shown in Figures 5e and 5f (Section 2.3)."

In L541 – L542, it's my opinion, you should remove the word "potential."

Thank you for this point. Suggested changes are made throughout the text, to ensure that the term "predicted $N_d$" is used only when comparing against the direct observations of cloud droplet numbers (i.e., in the droplet closure section), while the term "potential $N_d$" is adopted in the rest of the paper.

7. L606 "Lower $N_d$ values are visible during nighttime due to the limited turbulence."
Turbulence is lower near a surface at night, however, turbulence is being prescribed in Fig. 7. The diurnal cycle is explained on L320 – L321: "$N_{aer}$ at WOP peaks in the evening, reaching up to ⁓104 cm⁻³ presumably because of BB emissions in the valley which seem to stop around midnight (Fig. 1a)."

Thank you for pointing this out. What is driving the potential $N_d$ in Figure 7 is, indeed, the amount of aerosol particles rather than the turbulence which is prescribed. This sentence is now corrected.

8. Why did the locations of the SEA points change moderately from Figure 9 (acp-2020-1036-manuscript-version3.pdf) compared to an earlier draft of Figure 9?

The SEA data points presented in Figure 9 are derived from Figure 6 in Kacarab et al. (2020). This figure illustrates $N_d^{lim}$ for several research flights as a function of the measured characteristic vertical velocities, which are then translated into $\sigma_w$ in our case ($\sigma_w = w^*/0.79$). The inset plot of this figure shows a sensitivity test of how $N_d^{lim}$ varies for polluted, intermediate and clean conditions for $w^*$ values between 0.1 and 0.6 ms⁻¹. In the earlier version of our manuscript we superimposed the results from this sensitivity analysis, whereas in the latest version we decided to include the actual $w^*$ measurements. The latter is more consistent with the SEUS data points (derived from Fig. 6 in Bougiatioti et al., 2020), which are also obtained from aircraft observations in cloud legs.

**References:**

Bougiatioti, A., Nenes, A., Lin, J. J., Brock, C. A., de Gouw, J. A., Liao, J., Middlebrook, A. M., and Welti, A.: Drivers of cloud droplet number variability in the summertime in the southeastern United States, Atmos. Chem. Phys., 20, 12163–12176, https://doi.org/10.5194/acp-20-12163-2020, 2020.

Kacarab, M., Thornhill, K. L., Dobracki, A., Howell, S. G., O'Brien, J. R., Freitag, S., Poellot, M. R., Wood, R., Zuidema, P., Redemann, J., and Nenes, A.: Biomass burning aerosol as a modulator of the droplet number in the southeast Atlantic region, Atmos. Chem. Phys., 20, 3029–3040, https://doi.org/10.5194/acp-20-3029-2020, 2020.